# A Survey on Foundations and Frontiers of Multimodal Agentic Frameworks: Techniques and Applications

**Neel Mokaria**[*,1], **Rishie Raj**[*,1], **Dheeraj Baiju**[*,2], **Xiaoqian Shen**[3], **Shraman Pramanick**[4],
**Kevin Qinghong Lin**[5], **Arda Senocak**[6], **Mike Zheng Shou**[7], **Philip Torr**[5], **Mohamed
Elhoseiny**[3], **Yapeng Tian**[8], **Ruohan Gao**[1], **Salman Khan**[9], **Sayan Nag**[10], **Sanjoy Chowdhury**[#,1],
**Dinesh Manocha**[1]

[1] *University of Maryland, College Park,* [2] *IISc Bangalore,* [3] *KAUST,* [4] *Johns Hopkins University,*
[5] *University of Oxford,* [6] *Ulsan National Institute of Science and Technology,* [7] *National University of Singapore,*
[8] *University of Texas at Dallas,* [9] *MBZUAI,* [10] *University of Toronto*
[*] *Core contributors*      [#] *Core advisor*

*Correspondence to:  {nmokaria, rraj27, sanjoyc}@umd.edu, dheerajbaiju@iisc.ac.in*
**Reviewed on OpenReview:** *https://openreview.net/forum?id=eaVoaI7f8v*

## Abstract

Advances in large language models (LLMs) have fueled a wave of research into agency: the ability to reason, plan, and act. This effort has produced agentic frameworks that orchestrate perception, memory, and decision-making around powerful LLM backbones. With the advent of large multimodal models (LMMs), these systems can process and integrate diverse modalities, including images, audio, and video, thereby improving their real-world applicability. Yet, while surveys of LLM-based agents exist, the role of multimodality in shaping agency has not been systematically examined in recent years. This survey fills the gap by analyzing the impact of multimodality across the core functional modules of the agentic framework: perception, reasoning, planning, memory, and action. Using this lens, we trace the evolution from text-centric agents to multimodal frameworks, examine how modalities are integrated through delegated, late-fusion, and early-fusion architectures, and assess the emergence of agentic behaviors enabled by grounded perception and multimodal reasoning. We organize existing work through a modality-centric taxonomy that links architectural design choices to agent capabilities. Moreover, we review multimodal agentic systems across various application domains, including Robotics, GUI & Web Navigation, Multimedia Content Generation & Editing, and Long-form Video Understanding & Retrieval. Beyond capabilities, we analyze performance across these settings and discuss efficiency-scalability trade-offs, including training and inference costs, latency, and deployment constraints. By focusing on the impact of multimodality in agentic design, we aim to identify key gaps and chart a roadmap toward robust and general-purpose intelligent systems.

# Contents

# 1 Introduction

The concept of agents predates the creation of artificial neural networks. They have historically been used to refer to computer systems capable of autonomy in complex environments (Wooldridge & Jennings, 1995). A more mature definition was provided by (Wooldridge, 1999), which characterized an intelligent agent not only by autonomy but also by flexibility: the ability to be reactive, proactive, and socially capable. To replicate this flexibility artificially, researchers have long sought to decompose intelligence into functional cognitive modules. Contemporary multimodal frameworks now mirror this cognitive architecture, as illustrated in Figure 1: employing an *orchestrator* for executive function and memory, a *perception* module for grounding sensory inputs (Mesulam, 1998), and an *action* interface that executes decisions through tool use or embodiment, reflecting the sensorimotor loop essential for learning (Lave & Wenger, 1991; Barsalou, 2008).

However, early AI systems attempted to implement these components in isolation. Symbolic or logic-based agents, such as CONGOLOG (Lespérance et al., 1995) and METATEM (Barringer et al., 1990), focus on deduction and theorem proving but struggle to remain tractable in complex environments. To address this, researchers proposed reactive agents based on hierarchical stimulus-response rules (Brooks, 1985), yet these lacked long-term planning capabilities. Even with the advent of deep learning, progress remained compartmentalized: computer vision models (Krizhevsky et al., 2012; He et al., 2015) focused solely on recognition, planning algorithms (Erol et al., 1994; Ghallab et al., 1998) operated in symbolic states without sensory grounding, and reinforcement learning agents (Mnih et al., 2013; Silver et al., 2016) learned policies without explicit reasoning. The rise of foundational models has finally led to the integration of these disjoint research areas. When augmented with perception modules and action interfaces, these models develop a complete cognitive loop, moving from static knowledge engines to interactive agents.

This integration was primarily driven by the emergence of Large Language Models (LLMs). These models possessed extensive open-world knowledge (Brown et al., 2020; Wei et al., 2022a) and demonstrated emergent reasoning abilities (Wei et al., 2022b) due to their large-scale unsupervised pre-training. These capabilities enable them to serve as a powerful alternative to symbolic and rule-based planning observed in earlier agents. At this point, it is essential to clearly distinguish between foundational models, such as LLMs, and end-to-end agents, which are the primary focus of this survey. A foundational model can act as the reasoning and perception component of a framework, while an E2E agent is the complete framework built around this model to impact agency. This class of LLM-based agents (Reiichiro Nakano et al., 2021; Yao et al., 2022b; Tanmay Gupta, 2022; Shen et al., 2023; Ahn et al., 2022) has demonstrated a wide range of agentic tasks, such as web browsing, image editing, and robotic manipulation. However, it was found that their greatest strength is also their greatest limitation. Although LLMs are effective tools for reasoning and planning in the language domain, a significant loss of information occurs when converting various input modalities into language (Rohrbach et al., 2018; Petryk et al., 2024).

This limitation in LLMs gave rise to *large multimodal models* (LMMs), which are essentially foundational models with multimodal understanding rather than text-only models. The initial approach was to augment LLMs with multimodal capabilities using modality-specific encoders (Radford et al., 2021b; Elizalde et al., 2022; Girdhar et al., 2023) to encode various modalities into the word embedding space of the LLM (Tsimpoukelli et al., 2021; Zhang et al., 2023b; Alayrac et al., 2022; Li et al., 2023a; Bai et al., 2025b). We refer to such adapter-style models as having *late-fusion* multimodal perception. Examples of representative agentic frameworks that use this class of foundational models as their backbone include LLaVA-Plus (Liu et al., 2024), MiniGPT-4 (Zhu et al., 2023), PaLM-E (Danny Driess et al., 2023), and Magma (Yang et al., 2025). Subsequent approaches tried to integrate multimodal perception into the foundational model rather than encoding it externally and independently. This gave rise to end-to-end multimodal models such as (OpenAI, 2023; 2024; Gemini Team, 2024; Bai et al., 2024; Xu et al., 2025a; Chen et al., 2025c). The idea behind this approach was that natively multimodal models enabled true cross-attention between modalities for a deeper, more organic fusion whereas adapter models tried to externally fuse embeddings from the final projection layers of the unimodal encoders. This leads to stronger perception and reasoning in end-to-end models. We refer to this as *early-fusion* multimodal perception. We are starting to see the use of such models as the backbone of several agentic frameworks, particularly those that operate in complex digital and physical

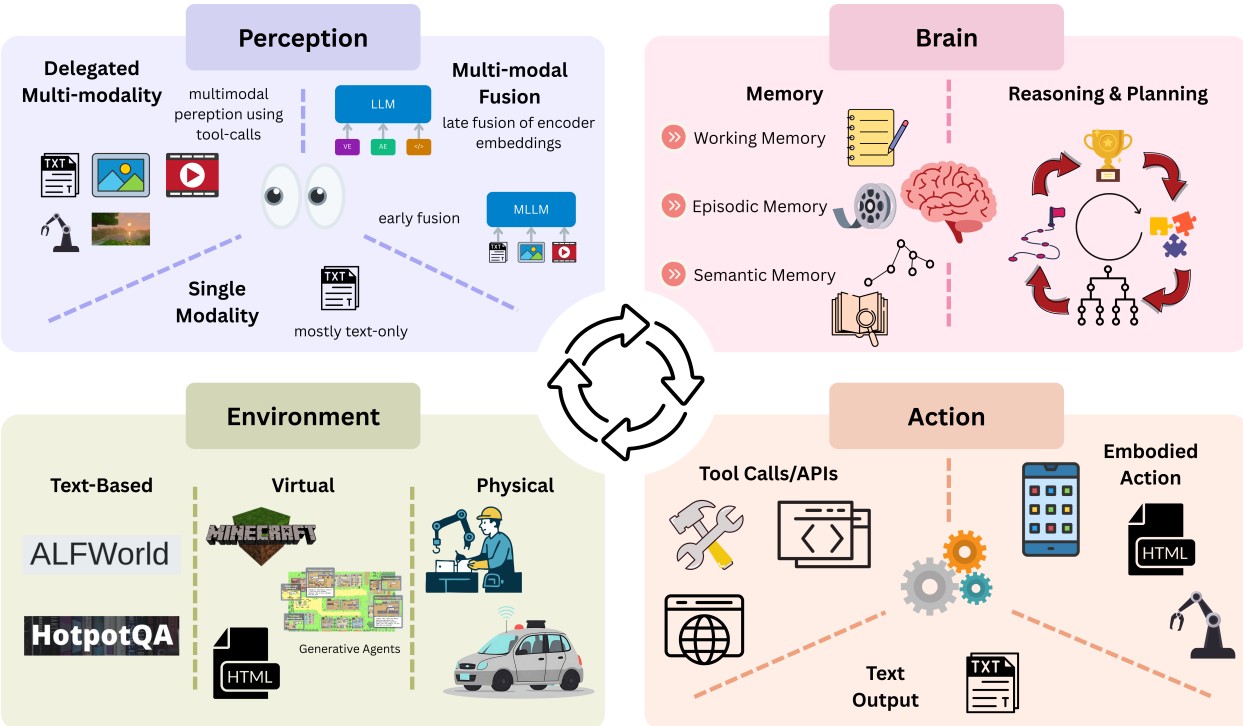

Figure 1: **Overall Agentic Architecture:** This figure depicts the three-component architecture inspired by cognitive principles discussed above. The framework comprises three interconnected modules: the *Perception* module processes various input modalities to inform the agent; the *Orchestrator* plans goals and stores interaction experiences; and the *Action* module interacts with the environment to achieve assigned objectives. The flow of signals between these components forms the cognitive loop that enables agentic behavior.

environments such as web/UI navigation (He et al., 2024a;b; Zheng et al., 2024) and robotics (Hu et al., 2023b; Wake et al., 2024). Both these fusion strategies have been explained in detail in Figure 2.

Based on this evolution, we establish a clear definition for the systems surveyed in this paper. We define a multimodal agent as a framework that utilizes an LMM as its core backbone to perceive, reason, and plan across multiple modalities simultaneously. This is the primary component of the larger cognitive loop of perceive → reason → remember → plan → act that the agent has to follow. This definition stands in direct contrast to earlier LLM-based agents, which are limited to reasoning over text-only inputs and suffer from information loss when converting other modalities into language. A true multimodal agent, by our definition, natively ingests and reasons over rich, multimodal representations to generate complex, grounded plans. These plans are then used to take actions, often by orchestrating a set of specialized tools, to interact with complex digital or physical environments. This category is further divided into agents built on late-fusion (adapter-style) LMMs and those that leverage the deeper, cross-modal understanding of early-fusion (end-to-end) LMMs. While the former has the advantage of being lightweight and faster for inference, the latter is better suited to fine-grained understanding and reasoning tasks.

**Note:** While the term *Agentic Framework* might be interpreted as an engineered system built around a base LLM, modern MLLMs are capable of agentic tasks by themselves. Therefore, in this paper, we use the term above to refer to all systems capable of agency.

Several surveys (Wu et al., 2024c; Huang et al., 2024; Xi et al., 2023; Chen et al., 2024b; Wang et al., 2024d; Ferrag et al., 2025; Zheng et al., 2025) have investigated agentic frameworks from different lenses, including architecture, reasoning, planning, continual learning, application, and evaluation. (Wang et al., 2024d) proposed a unified framework for autonomous agents comprising four key modules: profiling, memory, planning, and action. Similarly, (Xi et al., 2023) categorizes agentic architecture by separating brain, perception,

and action modules, inspired by cognitive science. Focusing on systems that enable continuous adaptation, surveys such as (Zheng et al., 2025) first discussed lifelong learning methods for LLMs across 12 scenarios involving internal and external knowledge of the agent. Surveys more aligned with multimodal agents (Wu et al., 2024c) chart the landscape of large multimodal agents, detailing architectures that fuse perception modules with language reasoning and action execution, and cataloging applications in web/OS automation, vision-language interaction, and embodied control. In the domain of multi-agent systems (MAS), (Guo et al., 2024) traced the evolution from single-agent LLM reasoning to collaborative MAS, detailing how profiling and communication protocols enhance collective problem-solving.

Despite there being a wide variety of surveys in this space, contemporary survey works primarily have two limitations: *(i)* most surveys only discuss LLM-based agentic frameworks, and *(ii)* none of them discuss the evolution of multimodality in agentic frameworks and how it has impacted their architecture. Existing surveys either focus narrowly on specific domains such as embodied AI (Ma et al., 2024b), web agents (Ning et al., 2025), and tool orchestration (Xu et al., 2025b) or treat multimodality as a peripheral feature rather than a central organizing principle (Wu et al., 2024c; Xi et al., 2023; Huang et al., 2024; Ferrag et al., 2025). Furthermore, there is a lack of unified analysis that connects foundational architectural choices, such as modality fusion strategies and memory mechanisms, with practical considerations, including efficiency and capability trade-offs, safety vulnerabilities, and evaluation methodologies. This survey addresses these gaps by providing a systematic examination of how multimodality has transformed agentic frameworks across their entire architectural stack, from perception and reasoning to memory and action.

The major contributions of our survey can be summarized as follows:

1. **Modality-Based Taxonomy and Architectural Analysis:** We provide a detailed analysis of how the introduction of multimodality has impacted the architecture of agentic systems. This includes a new taxonomy centered on the impact of multimodal integration across perception, reasoning, planning, memory, and action modules.

2. **Classification of Perception Fusion Strategies:** We categorize and formalize the evolution of perception into three distinct strategies: Delegated, Late-fusion, and Early-fusion perception. Our analysis demonstrates that frameworks that fuse multimodal information generally outperform those relying on language as an intermediate representation.

3. **Comprehensive Cross-Domain Performance Evaluation:** We correlate specific architectural choices with quantitative performance gains across four primary application domains: robotics and physical embodiment, GUI and web navigation, multimedia content generation, and long-form video understanding

4. **Efficiency and Scalability Trade-off Mapping:** We offer a detailed comparison of fusion strategies based on resource usage, task completion time, and deployment costs. This analysis identifies a critical roadmap for future research, highlighting the economic and latency advantages of fine-tuned, domain-specific architectures over proprietary, token-heavy APIs.

## 2 Foundations of Agentic Framework

As discussed earlier, modern agentic frameworks follow the fundamental cognitive loop (Figure 1) established in cognitive science, with foundational models as a central component of that loop. In this section, we will cover the fundamental components of the agentic framework and discuss the roles they play in helping an agent achieve its goals. We will also briefly examine the modalities in which these agents primarily operate, along with their applications and use cases.

### 2.1 Orchestrator

The orchestrator serves as the central decision-making unit of an agentic framework, responsible for reasoning, planning, and memory management. This design directly reflects cognitive architectures where executive functions coordinate lower-level processes to achieve goals. In human cognition, the prefrontal cortex acts as an executive controller that manipulates internal representations to draw inferences, decomposes complex goals into executable subtasks, and maintains both short-term working memory and long-term knowledge

stores. Early symbolic AI systems attempted to replicate these cognitive functions through logic-based reasoning engines and hierarchical task networks (Erol et al., 1994; Ghallab et al., 1998), but remained brittle due to their reliance on hand-crafted rules and inability to handle uncertainty. The paradigm shift came with the rise of foundation models (Bommasani et al., 2022), particularly large language models that demonstrated emergent cognitive capabilities through self-supervised learning at scale. The first generation of foundation models like GPT-3 (Brown et al., 2020) were text-only LLMs whose pre-training on massive text corpora provided them with several emergent capabilities (Wei et al., 2022b): (1) few-shot learning, enabling adaptation to new tasks from minimal examples; (2) multi-step reasoning, allowing decomposition of complex problems into logical chains; and (3) instruction following, permitting natural language control of agent behavior. These capabilities directly parallel the cognitive functions of human executive control, marking the transition from brittle symbolic systems to flexible, learning-based orchestrators.

The orchestrator's three core functions each draw inspiration from distinct aspects of human cognition. *Reasoning* mirrors the process of manipulating internal representations to draw inferences, which inspired Chain-of-Thought prompting (Wei et al., 2022c) and Tree-of-Thoughts search (Shunyu Yao et al., 2023) in LLMs. *Planning* reflects the hierarchical decomposition of goals into executable subtasks, as observed in cognitive architectures, leading to tool-orchestration (Qin et al., 2024) and meta-programming (Hong et al., 2023) frameworks in agentic systems. *Memory* encompasses the tripartite structure identified in neuroscience: working memory for augmenting the agent's context window (Zhu et al., 2024; Zihang Dai et al., 2019; Charles Packer et al., 2023), episodic memory for storing and reflecting on past experiences (Shinn et al., 2023b; Park et al., 2023; Wang et al., 2023a), and semantic memory for structured knowledge retrieval (Patrick Lewis et al., 2020; Speer et al., 2018). Together, these functions enable the orchestrator to move beyond reactive stimulus-response patterns toward goal-directed, adaptive behavior that can persist across extended interactions.

### 2.1.1   Reasoning

One of the emergent capabilities of LLMs is to perform reasonably well on tasks for which they were not explicitly trained, using in-context learning from natural-language prompts. This ability to learn *in* context led to the emergence of *reasoning* in LLMs, first showcased by Chain-of-Thought (CoT) (Wei et al., 2022c). This approach unlocked new capabilities in arithmetic, commonsense, and symbolic reasoning by guiding the model with step-by-step reasoning exemplars. Building on this, (Kojima et al., 2023) later shows that LLMs may contain stepwise reasoning capabilities as part of their pre-trained knowledge and may not require in-context exemplars. A simple trigger such as "Let's think step by step" can induce stepwise reasoning in a zero-shot setting, achieving performance comparable to few-shot CoT. This kind of reasoning ability was also shown to improve drastically with model size. Further improvements were made by (Xuezhi Wang† et al., 2023), which replaces greedy decoding with sampling multiple reasoning paths and selecting the most consistent answer.

### 2.1.2   Planning

The ability to reason directly leads to the ability to generate stepwise plans for accomplishing a task. The LLM uses its world knowledge and reasoning capabilities to decompose complex, high-level tasks into simpler, low-level subtasks. This can be seen in prompting strategies such as (Wang et al., 2023b) and in agentic frameworks such as (Yao et al., 2022b; Xu et al., 2023; Shunyu Yao et al., 2023). In these approaches, the model first decomposes a task, then interleaves these planning steps with observations that may be knowledge retrieval or environment feedback, and finally revises the plan as this evidence arrives. Various approaches have been proposed to improve the performance of these planning strategies, such as delegating mathematical computation to interpreters by generating the plan as executable code (Chen et al., 2023; Gao et al., 2023c). (Dwivedi-Yu et al., 2023) takes task delegation and tool calls further by training an LLM to use API/tool calls to solve tasks. For embodied tasks, frameworks such as (Ahn et al., 2022) ground plan generation in the real world through physical affordances and value models.

### 2.1.3   Memory

For LLMs to build knowledge and develop competence, they require augmentation with memory. This requires generating and storing traces that capture previous inputs and experiences, building a *persistent state* that can carry beyond the context window, and a retrieval-augmented knowledge database that can be

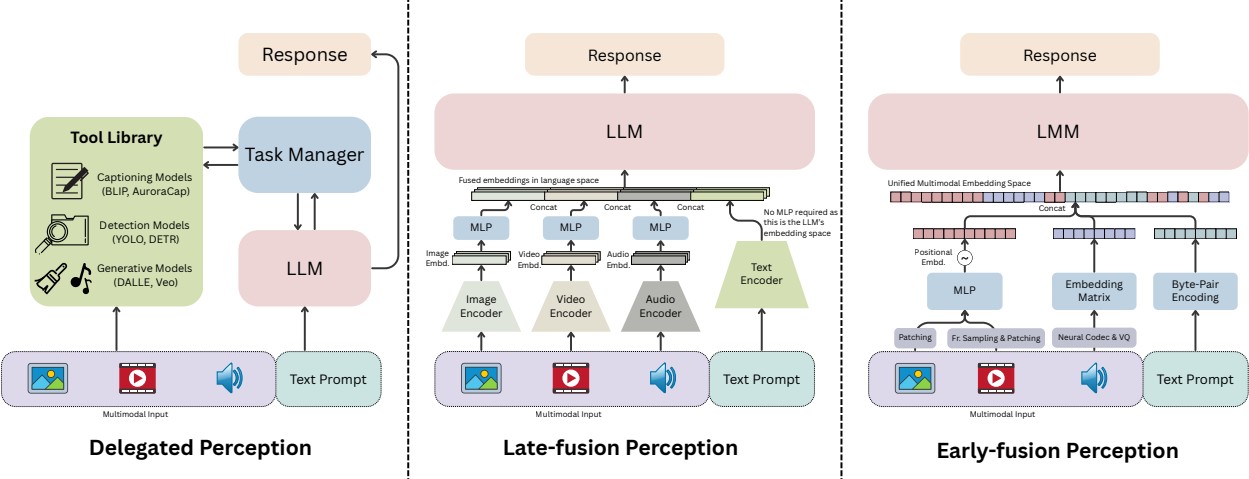

Figure 2: **Types of Perception Fusion:** This figure shows the three different fusion techniques used to merge information from different modalities. We have used these three categories to distinguish various frameworks in our survey. *Delegated Perception* uses tool calls to read various modalities and transform the information into natural language for LLM understanding. *Late-fusion Perception* uses modality-specific encoders to transform different modalities into the language embedding space, requiring domain-specific training. Finally, we have *Early-fusion Perception*, which natively uses raw multimodal inputs for perception.

queried by the LLM on demand. (Shinn et al., 2023b) is a representative work on episodic memory-based framework designs, in which the agent stores short reflections after each episode and reuses them for the next action. On the other hand, more deterministic designs, such as (Hu et al., 2023a), maintain an external database that can be queried by the LLM via SQL queries. There are also frameworks such as (Zhong et al., 2023) that support a long-term memory framework, reinforced by a strengthening-and-forgetting principle. Interactive memory frameworks, such as (Huang et al., 2023c), have also been explored, allowing users to control an LLM's memory state and objects through an interactive interface. However, this idea never gained widespread adoption. Finally, RAG (Patrick Lewis et al., 2020) has become a popular memory framework for LLMs, as it enables them to retrieve specific parts of their past experiences conditionally on user preferences. This allows the LLM to focus not on the entire memory state but rather on retrieving specific parts required for the current planning action. (Zhang et al., 2025b), in particular, enables hierarchical-style retrieval of multimodal memory objects stored in heterogeneous memory states, such as vectors, graphs, and web databases.

## 2.2 Perception

The perception module is the key component that imparts multimodal capabilities to agentic frameworks. It converts raw multimodal inputs into grounded abstractions, enabling the orchestrator to reason and develop a plan to execute the user-provided instructions. A multimodal perception unit also enables memory to store multimodal states, thereby supporting multimodal reasoning and planning. The hierarchical processing observed in biological sensory systems (Hubel & Wiesel, 1962; Mesulam, 1998) has directly inspired both convolutional neural networks (Krizhevsky et al., 2012) for vision-based learning and cross-attention mechanisms in transformer-based (Vaswani et al., 2017) multimodal architectures. The evolution of perception modules reflects a progression from delegated tool-based processing to native multimodal understanding:

**Delegated Perception:** The most popular early form was delegated perception (Tanmay Gupta, 2022; Yang et al., 2023; Yujia Li et al., 2022; Shen et al., 2023; Chowdhury et al., 2025d; Galougah et al., 2025a; Chowdhury et al., 2026; 2025b;c), where the orchestrator (text-only LLM) calls multimodal perception tools to process non-textual inputs. These tools can be dedicated action models, such as CLIP (Radford et al., 2021b) for vision-language alignment, CLAP (Elizalde et al., 2022) for audio understanding, SAM (Kirillov et al., 2023) for image segmentation, and Whisper (Radford et al., 2022) for speech recognition, or complete foundation models, such as LLaVA (Zhang et al., 2023b) and InstructBLIP (Li et al., 2023b). While this

approach offers modularity and specialization, it suffers from information loss during the translation of multimodal inputs into text-based representations (Rohrbach et al., 2018; Petryk et al., 2024).

**Late-fusion Multimodal Perception:** To solve the information loss problem in the above approach, researchers started using modality fusion techniques to merge input embeddings into the embedding space of the LLM. They adopted strategies from (Zhang et al., 2023b; Alayrac et al., 2022; Li et al., 2023a; Chowdhury et al., 2024a;b; 2025a) to achieve this multimodal fusion. This gave rise to agentic frameworks like (Furuta et al., 2023; Hong et al., 2024; You et al., 2024; Ichter et al., 2023; Danny Driess et al., 2023; Kim et al., 2024) which fused multimodal inputs like images, video, and audio into a base LLM and were fine-tuned on domain and task-specific data. A major advantage of these models was that they were open source, because of which they could be locally deployed and distilled into smaller models for faster inference when fine-tuned for a specific application.

**Native Multimodal Processing:** With the rise of multimodal models (OpenAI, 2024; Gemini Team, 2024; Xu et al., 2025a; Meta AI, 2024; Chowdhury et al., 2023b;a; 2021a;b) with strong reasoning and agentic capabilities, the field is observing a fundamental shift towards *natively multimodal* perception, where input processing is performed directly by the foundation model itself rather than delegated to external tools. These models integrate perception through early fusion mechanisms that enable direct cross-modal attention, preventing significant information loss and maintaining access to fine-grained details such as spatial relationships in images, temporal dynamics in video, or prosodic features in speech. This shift from tool orchestration to native multimodality has proven critical for tasks requiring precise visual grounding, such as GUI navigation (Koh et al., 2024; Lin et al., 2025) and embodied control (Ichter et al., 2023; Danny Driess et al., 2023).

**Note:** Perception strategies transition from *delegated* (modular tool-calls with text-based reasoning (Chowdhury & Manocha, 2026)) towards *late-fusion* (embedding projections of encoder features) and *early-fusion* (native processing of raw multimodal tokens), shifting from textual abstractions of inputs to deep, unified multimodal reasoning.

## 2.3 Action

With a stepwise plan in place, the agent turns to the action module to perform tasks and interact with the environment, which may also involve receiving feedback. Following the embodied cognition principles outlined in Section 1, where intelligence emerges from real-world experiences and interactions (Lave & Wenger, 1991; Pavlov, 2010), modern action modules close the perception-action loop by grounding plans in executable operations. In the context of multimodal agents, *action* typically involves tools/API calls, UI/webpage navigation, code execution, or physical embodiment, along with updating memory with the resulting observations. Tool-calling agents such as (Yao et al., 2022b; Dwivedi-Yu et al., 2023; Xu et al., 2023) use a text-only LLM as a base to generate appropriate tool calls from user instructions and the task to be accomplished. Both input processing and output generation are components of the agent's action module. A more generalist agent is a code-generating agent (Chen et al., 2023; Liang et al., 2023; Yujia Li et al., 2022; Tanmay Gupta, 2022) that executes scripts to perform actions. They can be used for computations, data transformation, and the orchestration of toolchains. Recent agents have been more focused on embodied actions like Web/UI navigation (Reiichiro Nakano et al., 2021; Zhang et al., 2023a; Zheng et al., 2024) and robot manipulation (Ichter et al., 2023; Danny Driess et al., 2023; Kim et al., 2024; Ahn et al., 2022).

## 2.4 Modalities

Agentic frameworks operate across diverse environments, ranging from digital interfaces to physical embodiment. Depending on the foundation model, these modalities can be processed via delegated tools or natively through embedding fusion. Below, we summarize representative frameworks and tasks for each modality.

**Text** Text serves as the foundational modality for agentic systems, acting as the primary medium for reasoning, code generation, and tool orchestration. Early frameworks such as WebGPT (Reiichiro Nakano et al., 2021) and ReAct (Yao et al., 2022b) demonstrated that agents could interact with external environments solely through language, either by parsing HTML to browse the web or by interleaving reasoning traces with

textual actions. This text-based paradigm also extends to tool use, where ToolFormer (Dwivedi-Yu et al., 2023) enables models to learn to invoke APIs via string patterns, and to complex software development, where multi-agent frameworks like MetaGPT (Hong et al., 2023) utilize natural language protocols to orchestrate coding tasks among specialized agent roles. Additionally, text often functions as a crucial abstraction layer for systems operating in digital environments. For example, agents such as AutoWebGLM (Lai et al., 2024), WebAgent (Gur et al., 2023), and AutoDroid (Wen et al., 2024) bridge the gap between visual interfaces and language models by converting rich GUIs into simplified HTML or XML representations, thereby enabling efficient navigation and interaction without requiring heavy visual encoders.

**Images** Images serve as the primary modality alongside text for providing rich context in perception and reasoning. Early frameworks leveraged LLMs to orchestrate specialized vision tools; for instance, VisProg (Tanmay Gupta, 2022) and MM-ReAct (Yang et al., 2023) employed code generation and prompt engineering to compose tools such as captioners and detectors for visual understanding and editing tasks. ViperGPT (Yujia Li et al., 2022) extended this by synthesizing Python programs to query vision APIs for complex reasoning. Bridging the gap between tool use and end-to-end learning, frameworks like LLaVA-Plus (Liu et al., 2024) were trained specifically to invoke visual tools, enabling more effective image manipulation and detailed analysis. Beyond static analysis, the image modality is fundamental to embodied agents and GUI navigation, in which models must perceive and interact with visual environments via camera feeds or screenshots. Grounding high-level plans in physical affordances, SayCan (Ahn et al., 2022) uses value functions derived from visual observations, while Vision-Language-Action (VLA) models like RT-2 (Ichter et al., 2023) and OpenVLA (Kim et al., 2024) directly output robot actions from visual inputs. Similarly, in the digital domain, frameworks such as AppAgent (Zhang et al., 2023a), SeeAct (Zheng et al., 2024), Agentic-DRS (Nag et al., 2025), and WebVoyager (He et al., 2024a) leverage MLLMs' visual understanding to perceive screen elements and execute complex workflows on web and mobile interfaces.

**Video** Video introduces temporal dependencies and high computational costs, requiring specialized strategies for long-form understanding. VideoAgent (Wang et al., 2024e) addresses this by using iterative frame retrieval to answer questions about hour-long videos efficiently. LLoVi (Zhang et al., 2024) employs a two-stage decomposition, generating textual descriptions of clips for temporal reasoning without heavy training. For more complex reasoning, VideoMind (Liu et al., 2025) utilizes role-specialized adapters to switch between planning and verification modes, while VLog (Lin & Shou, 2025) treats video narrations as vocabulary units to enable fast, generative retrieval of semantic events. LVAgent (Chen et al., 2025a) enables multi-round dynamic collaboration of agents in long video understanding. Vgent (Shen et al., 2025) represents videos as graphs and introduces structured reasoning with information aggregation to enhance long-video analysis.

**Audio** Agentic frameworks in the audio domain focus on both fine-grained reasoning and generation. WavCraft (Liang et al., 2024) and WavJourney (Liu et al., 2023b) leverage LLMs to plan and execute audio editing and creation pipelines using external tools. For high-fidelity generation, SonicRAG (Guo & Tai, 2025) introduces a retrieval-augmented generation framework to condition sound effect synthesis on database retrieval. More recently, multi-agent systems such as ReelWave (Wang et al., 2025d) have been developed to handle complex tasks, including video-to-audio synthesis, in which agents collaboratively generate and synchronize audio tracks with visual content. The tool-calling framework has been demonstrated in (Wijngaard et al., 2025) that coordinates multiple audio-language models as specialized tools via an LLM orchestrator. There are also multi-agent systems, such as (Rong et al., 2025), that decouple perception from cognition by recasting audio reasoning as a text-based understanding task.

> **Note:** Text and audio transcripts can provide a stable semantic abstraction for processing these modalities with *minimal contextual loss*. However, dense modalities like images and video require native multimodal processing to capture spatial and temporal details that are *lost during text conversion*.

## 3 Taxonomy

Having established the fundamental architecture of agentic systems and the specific modalities in which they operate, we now examine how their core components have evolved to accommodate multimodality. The transition from text-only agents to multimodal systems is not merely an additive process of attaching

perception modules. It requires fundamental architectural adaptations to ensure that information is not lost during translation among different sensory inputs, reasoning processes, and executable actions. We will discuss in detail the terms we use to analyze these systems throughout the remainder of the paper, to understand agentic frameworks from a multimodal perspective.

### 3.1 Multimodal Perception Strategies

The first challenge in a multimodal agentic framework is to convert the raw sensory data into a format that the orchestrator can process. We categorize perception strategies according to the degree of integration between the sensory encoder and the reasoning backbone. Broadly, multimodal inputs can either be converted into a model-understandable modality or be processed natively. Even within native perception, the orchestrator either receives encoded inputs (via independent multimodal encoders) or uses raw input tokens directly. As research on agentic systems has evolved, we have observed a range of frameworks across these three categories, as shown in Figure 4. We discuss each of these approaches in detail with their associated exemplars.

#### 3.1.1 Delegated Perception (via Tool Calling)

The earliest and most modular approach involves treating perception as an external tool call. In this paradigm, a text-only LLM acts as a controller that orchestrates specialized off-the-shelf vision or audio models. The LLM generates a plan that identifies data needs, invokes a specialized API, and ingests the resulting text. These experts convert non-text modalities into natural-language descriptions, such as captions, bounding boxes, or transcripts, which are then fed back into the LLM. Frameworks such as HuggingGPT (Shen et al., 2023) and Visual ChatGPT (Wu et al., 2023) exemplify this strategy, dynamically selecting models from libraries to process incoming sensory data. MM-ReAct (Yang et al., 2023), VISPROG (Tanmay Gupta, 2022), and CLOVA (Gao et al., 2024b) extend this by enabling the LLM to actively decide when to delegate specific vision tasks, such as face recognition, to external services, such as Azure Cognitive Services, rather than attempting them natively. Similarly, Chameleon (Lu et al., 2023) refines this by composing a plug-and-play inventory of tools for mixed-modal reasoning.

#### 3.1.2 Late-Fusion Perception

This mechanism typically involves a frozen visual encoder, such as CLIP (Radford et al., 2021b) or a Vision Transformer (ViT) (Dosovitskiy, 2020), which extracts features that are then passed through a trainable projection layer. EgoVLP (Lin et al., 2022) introduced egocentric video-text contrastive pretraining on egocentric videos, and EgoVLPv2 (Pramanick et al., 2023) strengthened the representations through cross-modal fusion in the backbone, yielding encoders well suited to serve as the perceptual for downstream late-fusion task. Flamingo (Alayrac et al., 2022) established the foundational blueprint for this approach with its Perceiver Resampler, which compresses diverse visual features into a fixed number of visual tokens, allowing a frozen LLM to reason over interleaved image and text sequences without extensive retraining. This architecture has evolved to support complex agentic behaviors in LLaVA-Plus (Liu et al., 2024), which extends the base architecture by training the model to actively invoke external visual tools directly from pixel inputs, while Magma (Yang et al., 2025) enhances this perception style with spatial and temporal awareness by training adapters to process Set-of-Marks prompts. Similarly, video-focused late-fusion models such as LongVLM (Weng et al., 2024), Video-XL (Shu et al., 2024), and MovieChat (Song et al., 2024) apply the same principle using frozen video encoders with simple projectors. This late-fusion strategy, which injects visual tokens into a language backbone, has proven sufficiently robust to serve as the perceptual engine for embodied controllers such as RT-2 (Ichter et al., 2023) and OpenVLA (Kim et al., 2024), which translate these visual tokens directly into motor commands. PaLM-E (Danny Driess et al., 2023) follows a similar late-fusion scheme for robotic perception. A comparable design is used in GUI/navigation agents such as Pix2Act (Shaw et al., 2023), WebGUM (Furuta et al., 2023), CogAgent (Hong et al., 2024), and Ferret-UI (You et al., 2024), which also rely on frozen encoders with lightweight projection layers to map interface images into the LLM space.

#### 3.1.3 Early-Fusion Perception

The current frontier of perception involves native, end-to-end integration. Early-fusion models process all modalities through a unified architecture, often a single transformer backbone trained from scratch or via heavy adaptation on mixed-modal sequences. In this mechanism, inputs from different modalities, whether

text, pixel patches, or audio waves, are tokenized into a unified vocabulary and processed via shared self-attention layers. This enables native cross-attention, capturing subtle nuances lost in translation, such as the tone of voice in audio or temporal micro-expressions in video. Models such as GPT-4o (OpenAI, 2024) and Gemini 1.5 (Gemini Team, 2024) employ native multimodal attention to handle interleaved text, audio, and video inputs seamlessly, setting the standard for this paradigm. This approach has been further democratized by architectures such as Chameleon (FAIR, 2024), which tokenizes images and text into a single discrete stream and applies standard language modeling techniques to multimodal data without separate encoders. Fuyu-8B further simplifies this by directly projecting raw image patches into the transformer, treating pixels almost exactly like text characters. Similarly, ViLa (Chen et al., 2024c) applies this early-fusion principle by embedding image patches directly into the shared token space. The field is now advancing toward architectures that unify not just understanding but also generation within the perception loop; Transfusion (Zhou et al., 2024a) and Show-o (Xie et al., 2024) demonstrate single transformers that can simultaneously predict next-text tokens and diffuse continuous image representations. To address the engineering challenges of this unification, Janus-Pro (Chen et al., 2025b) introduces a decoupled visual encoding strategy within a unified autoregressive transformer, resolving the trade-off between the granularity required for visual generation and the abstraction required for high-level visual understanding.

### 3.2 Multimodal Reasoning and Planning

This stage constitutes the *orchestrator* of the agentic framework, in which multimodal inputs are transformed into a logical sequence of sub-goals. Unlike text-only reasoning and planning, multimodal planning requires the orchestrator to maintain a coherent internal state that accounts for spatial, temporal, and semantic relationships across different modalities. We categorize these reasoning strategies by the extent to which the inference process engages with textual information, progressing from purely symbolic logic to deeply grounded cross-modal reasoning.

#### 3.2.1 Language-based Reasoning

In this approach, reasoning occurs entirely within the language space. Even if the input is an image, the reasoning process operates on a textual abstraction of that image generated by the perception module. The agent converts observations into a *thought trace* in text and uses standard logical inference to break down tasks into Thought → Action → Observation loops. This paradigm is rooted in foundational prompting strategies that first unlocked systematic cognitive-like behaviors in LLMs, such as Chain-of-Thought (Wei et al., 2022c), which demonstrated that eliciting step-wise reasoning traces allows models to solve multi-hop problems through logical decomposition. This capability was further expanded by Tree-of-Thoughts (Shunyu Yao et al., 2023), a framework that enables the orchestrator to explore multiple potential solution branches and backtrack upon detecting likely failures, effectively transforming the model into a deliberate solver capable of non-linear planning. The ReAct (Yao et al., 2022b) framework relies on these strategies to interleave reasoning with action execution. Reflexion (Shinn et al., 2023b) applies this to memory, converting failed multimodal trials into textual *lessons* that the agent uses in future episodes to avoid repeating errors.

#### 3.2.2 Visually-grounded Reasoning

To overcome the abstraction inherent in language, visually grounded reasoning strategies directly reference visual representations during inference. The reasoning process incorporates spatial coordinates, visual pointers, or Set-of-Marks (SoM) overlays to anchor logic to specific pixels or regions. This is critical for tasks involving spatial relationships, such as digital navigation and physical manipulation. In the digital domain, frameworks such as SeeAct (Zheng et al., 2024) and WebVoyager (He et al., 2024a) reason about specific screen coordinates and the functionality of web elements by overlaying numbered tags or bounding boxes directly on screenshots to guide the orchestrator's decision-making. Similarly, in robotics, vision-language-action models such as RT-2 (Ichter et al., 2023) and OpenVLA (Kim et al., 2024) achieve visual grounding by learning to map web-scale semantic knowledge directly to motor control through unified tokenization, whereas Magma (Yang et al., 2025) is designed to process SoM/ToM-based prompts for grounding. In the video domain, agentic frameworks such as VideoAgent (Wang et al., 2024e) maintain active reasoning chains to track temporal dynamics in videos, yielding state-of-the-art performance on long-form reasoning benchmarks, including EgoSchema (Zhuang et al., 2023).

### 3.2.3  Cross-modal Reasoning

Cross-modal reasoning represents the most advanced form of agentic inference, in which the orchestrator must simultaneously integrate discrete, often asynchronous, sensory inputs to resolve environmental ambiguities or conflicting instructions. This capability is fundamentally anchored in architectures such as ImageBind (Girdhar et al., 2023), which binds six distinct modalities into a unified embedding space, enabling agents to infer connections between signals without explicit pairwise training. Building on this, frameworks such as PandaGPT (Su et al., 2023) leverage these joint representations to execute complex instructions, such as locating an object based solely on its acoustic signature. This paradigm has been further expanded by *any-to-any* models such as NExT-GPT (Wu et al., 2024a), which use an LLM core to connect modality-specific encoders and decoders, enabling the agent to reason across and generate any combination of text, images, videos, and audio. Similarly, CoDi-2 (Tang et al., 2024) introduces a composable diffusion-based approach that enables the orchestrator to perform in-context multimodal reasoning and follow complex instructions to generate interleaved multimodal content with high semantic alignment. We also have agentic frameworks such as ReelWave (Wang et al., 2025d) and AudioAgent (Anonymous, 2024) that employ cross-modal reasoning to synchronize audio tracks with dynamic visual content, ensuring temporal and semantic alignment across heterogeneous media types.

### 3.3  Multimodal Memory Architecture

To function over long horizons, agents must retain information across three distinct cognitive timescales. Working Memory serves as the agent's active reasoning buffer, managing the immediate context and current task state to support real-time decision-making. Episodic Memory serves as a storage system for past experiences and interaction histories, enabling the agent to recall specific events, sequences, and outcomes from previous episodes. Semantic Memory houses general world knowledge and facts that are independent of specific experiences, providing the agent with a static base of information about the environment. The implementation of these distinct memory functions depends on whether the agent handles modalities in isolation or through a unified representation.

### 3.3.1  Modality-specific Memory

In this architecture, the agent maintains distinct storage systems for different data types, utilizing text-based indices or symbolic pointers to bridge the gap between modalities. Working memory in this paradigm typically remains text-centric, with visual or auditory inputs not held directly in the active context but instead converted into textual descriptions or log entries (e.g., "User uploaded image.jpg"), thereby keeping the heavy perceptual load outside the cognitive core. This approach is exemplified by frameworks such as HuggingGPT (Shen et al., 2023) and Visual ChatGPT (Wu et al., 2023), in which the agent's state is maintained as a JSON-like history of tool executions and file references. Episodic memory is similarly compartmentalized, with past experiences stored in structured databases that separate raw media files from their textual narratives. Retrieval is a two-step process: the agent first queries a text log to identify a relevant timestamp or frame ID, and then retrieves the associated visual data. VideoAgent (Wang et al., 2024e) implements this strategy for long-form video understanding, treating the video not as a continuous stream but as a database of discrete visual events linked by time. Semantic memory involves querying separate, domain-specific databases, such as a text knowledge graph for facts and a separate image retrieval system for visual examples, and manually aggregating the results. This *late-fusion* retrieval strategy is often adopted by systems to optimize speed, searching specialized text indices for facts while keeping heavy media assets in cold storage until explicitly requested.

### 3.3.2  Unified Memory

Unified architectures project information from all modalities into a shared semantic embedding space, thereby dissolving the boundaries between data types and enabling fluid cross-modal associations. Working memory in these systems becomes inherently multimodal, utilizing a context window that accepts interleaved sequences of text, image, audio, and video tokens. This allows models like Gemini 1.5 (Gemini Team, 2024) and GPT-4o (OpenAI, 2024) to process and attend to specific image patches or audio segments directly within the active reasoning buffer. Episodic memory leverages cross-modal vector stores in which experiences are stored as unified embeddings. This enables content-based retrieval, allowing an agent to retrieve

a past visual experience from a textual description or to retrieve a conversation from a visual cue. HM-RAG (Zhang et al., 2025b) explicitly structures this as a hierarchy, enabling agents to retrieve multimodal *memory objects* that encapsulate the holistic state of a past event rather than disjoint fragments. Finally, semantic memory is organized by concept rather than data format, binding different modalities to a fixed embedding space. Architectures such as ImageBind (Girdhar et al., 2023) serve as the foundation for this type of memory, creating a shared space in which arithmetic operations on concepts yield semantically consistent results. SonicRAG (Guo & Tai, 2025) extends this to the audio domain, allowing agents to retrieve sound effects from a semantic database to generate audio outputs that are contextually aligned with the narrative.

### 3.3.3 Temporal Context Management

Temporal context can grow as interaction states accumulate over time. This becomes an issue because maintaining a coherent task context becomes more costly as the number of steps, episodes, and stored experiences increases. There are several approaches to managing exploding temporal context, especially in multimodal interactions. A basic one is context compression, which is exemplified by Mobile-Agent-v2 (Wang et al., 2024b), where a dedicated planning agent continuously rewrites the full interaction history into a running text-based task-progress summary. Reflexion (Shinn et al., 2023b) applies an analogous strategy across episodes, where, at the end of each failed trial, the agent distills the failure into a short verbal reflection stored in episodic memory and prepends it to future episodes. AppAgent-v2 (Li et al., 2024) has transformed content management into a retrieval problem by building a static reference document during an offline exploration phase and accessing it via RAG at inference time. This removes the problem of session length as the context is now determined by the size of the retrieved chunk. GUI-Owl (Ye et al., 2025) imposes a hard limit on the context length by retaining only the most recent 1–3 screenshots of the interactions. The common limitation across all these strategies is that context compression leads to losses, which are more significant in the case of multimodal agentic systems as compared to text-based agents.

### 3.3.4 Bandwidth Management

While context management is important for controlling the real-world deployment of agent systems, the bandwidth of agentic interactions measures the amount of information flowing through the framework, as well as the orchestrator. A high bandwidth means the per-timestep information arriving is dense, hence this is another way in which the model context can fill up pretty quickly. To control the communication bandwidth, a common strategy that was pioneered by VideoAgent (Fan et al., 2024) is sparse selective retrieval, which maintains a CLIP embedding index over all video frames in RAM and retrieves an average of only 8.4 frames per query, achieving a 20x reduction in frames processed compared to dense sampling. LLoVi applies a simpler variant by sampling a sparse fixed set of frames, captioning them in parallel, and reasoning entirely over the resulting text, eliminating visual encoding cost for the LLM reasoning step. In a different approach, DoraemonGPT (Huang et al., 2025) converts continuous video streams into a spatio-temporal symbolic memory of object states and event timestamps, so the memory footprint scales with the number of semantically distinct events rather than raw frame count. VLog maps video content into a compact narration, reducing retrieval complexity from the number of frames to the vocabulary size and achieving a 10-20x speedup over baselines.

## 3.4 Multimodal Action Spaces

The transition from text-based to multimodal agents has fundamentally expanded the scope of executable actions, requiring new paradigms for how agents interact with their environments. While early agents were restricted to text generation or code synthesis, multimodal frameworks must now ground their decisions in diverse output modalities, ranging from pixel coordinates and motor-control signals to direct multimedia manipulation. This transition forces the agent not only to reason about what action to perform but also to translate the action into domain-specific, non-linguistic modalities required by the environment.

### 3.4.1 Language-driven Actions

In this paradigm, the agent's *action* is the generation of a structured text command, such as a JSON object, a Python function call, or an API request, which is then executed by an external interpreter. This decouples the agent's reasoning from the complications of physical or digital execution. ToolFormer (Dwivedi-Yu

et al., 2023) pioneered this approach by demonstrating that LLMs can learn to use tools via self-supervised learning. Gorilla (Patil et al., 2023) significantly advanced this by fine-tuning models on a massive corpus of API documentation, reducing hallucination and enabling precise argument generation for complex function calls. Moving beyond text-only tools, MLLM-Tool (Guo et al., 2025) explores how multimodal LLMs can be taught to utilize tools that accept or produce non-textual data, such as generating images or processing audio, thereby expanding the action space into the multimodal domain via language mediation. To scale these capabilities, ToolLLM (Qin et al., 2023) introduces a framework for instruction-tuning LLMs on the massive ToolBench dataset, enabling agents to master over 16,000 real-world RESTful APIs and generalize to unseen tools, a critical step towards open-ended agency.

### 3.4.2 Visually-Grounded Actions

To bridge the gap between semantic intent and pixel-level interaction, agents operating in digital and creative environments rely on visually-grounded actions that identify specific coordinates or visual elements within a scene. In the digital domain, frameworks such as AppAgent (Zhang et al., 2023a) enable agents to navigate smartphone applications by mimicking human gestures, such as taps and swipes, whereas CogAgent (Hong et al., 2024) specializes in high-resolution GUI perception to interact with dense, small-scale elements typical of desktop interfaces. These actions are often anchored in the Document Object Model (DOM), as in WebVoyager (He et al., 2024a) and Mobile-Agent (Wang et al., 2024c), which require precise localization to interact effectively with dynamic web content. Beyond digital navigation, visual grounding is essential for precise physical interaction, as we often find in robotics. As we explained in Section 3.2, Magma (Yang et al., 2025) employs a SoM strategy to anchor its reasoning to specific visual points. This enables precise manipulation by mapping instructions to the exact pixels representing an object. Similarly, VoxPoser (Huang et al., 2023b) synthesizes code to generate 3D value maps that guide robot trajectories based on the environment's spatial geometry.

### 3.4.3 Embodied Multimodal Actions

The most sophisticated action space involves physical interaction with the real world, requiring a tight integration between multimodal perception and continuous motor control. Unlike discrete coordinate clicks in the earlier case, embodied actions often require the orchestrator to output continuous values, such as joint velocities or gripper poses, or to invoke high-level motion primitives that operate on constant visual feedback. This paradigm is exemplified by Vision-Language-Action (VLA) models such as RT-2 (Ichter et al., 2023), in which robot actions are tokenized and generated directly by the foundation model, leveraging web-scale pre-training to generalize to novel physical tasks. OpenVLA (Kim et al., 2024) further democratizes this approach by providing open-source policies that can be fine-tuned for specific embodiments. Grounding these high-level plans often involves value functions, as seen in SayCan (Ahn et al., 2022), which evaluates the physical feasibility of a plan based on current visual observations. Other frameworks, such as PaLM-E (Danny Driess et al., 2023), inject multi-sensor data directly into the model's embedding space, enabling the agent to internalize world knowledge and maintain a coherent state throughout complex, real-world manipulation sequences.

### 3.5 Evaluation Methodologies

In order to compile a comprehensive list of strategies used across various agentic application domains, we conducted an audit of evaluation methodologies employed across all representative frameworks surveyed in our paper. We organize our findings in a Table 1, which classifies evaluation approaches into five categories: (1) environment-based task completion, (2) programmatic functional correctness, (3) deterministic automated metrics, (4) human evaluation, and (5) LLM-as-a-judge. While we were apprehensive about the preference of non-deterministic metrics, due to the nature of agentic task outputs, our analysis shows that LLM-as-a-Judge evaluation accounts for only ~5% of all surveyed frameworks, confined exclusively to the GUI and web navigation domain.

In the domain of robotics and physical embodiment, all representative frameworks (Ahn et al., 2022; Ichter et al., 2023; Kim et al., 2024; Octo Model Team et al., 2024) rely exclusively on task completion or success rate measured through real-time execution in simulated or physical environments; no LLM-based evaluator is involved. Similarly, long-form video understanding agents such as VideoAgent (Fan et al., 2024), LLoVi (Zhang et al., 2024), and VideoMind (Liu et al., 2025) uniformly employ deterministic evaluation

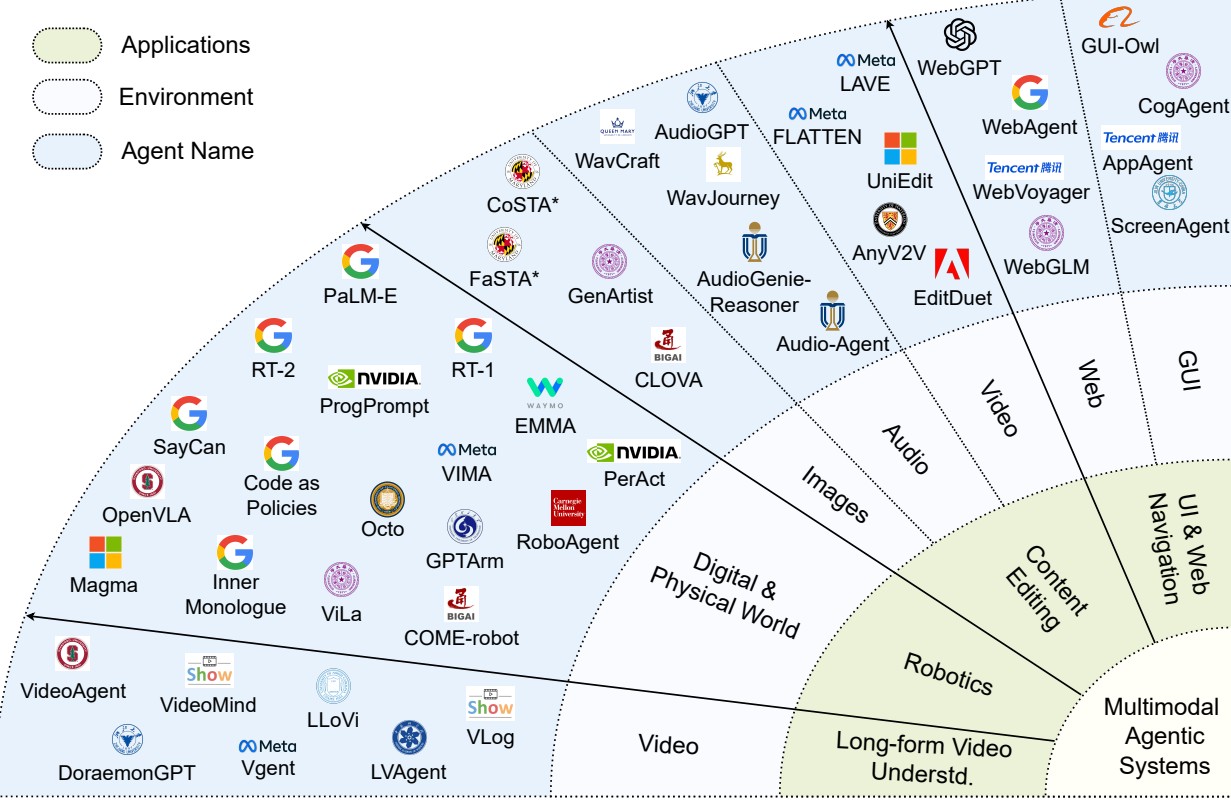

Figure 3: **Domain Classification:** This is a classification of representative agentic frameworks based on the *four* primary application areas we have discussed in the paper. They have been further classified according to the environments in which they operate. In the field of *Robotics* in particular, most agentic frameworks have been evaluated in both virtual and real-world physical environments.

metrics, including MCQ answer accuracy and temporal intersection-over-union (IoU). In multimedia content generation, frameworks such as GenArtist (Wang et al., 2024f), AudioGPT (Huang et al., 2023a), and WavCraft (Liang et al., 2024) rely on a combination of deterministic benchmarks (T2I-CompBench, FID, CLIP Score) and human evaluation studies. The only domain where LLM-based evaluation has been adopted is GUI and web navigation, and even there, its use is restricted. WebVoyager (He et al., 2024a) introduced an automatic evaluation protocol that uses GPT-4V to assess task trajectories, reporting 85.3% agreement with human evaluators. WebArena (Zhou et al., 2024b) employs GPT-4 fuzzy matching as one of several evaluation functions, applied specifically to information-seeking tasks where the format of agent responses is diverse and exact string matching is insufficient. The majority of WebArena's evaluation relies on programmatic state validators and exact/substring matching. Other GUI benchmarks evaluated in our surveyed frameworks, including VisualWebArena (Koh et al., 2024) and Mind2Web (Deng et al., 2023), predominantly employ programmatic functional correctness checks.

## 4 Application

We have discussed the architectural changes required in agentic systems to support multimodality. We now examine how agentic frameworks have evolved in their applications over the years. Figure 3 shows a classification of representative agentic frameworks from major application areas. We will start with the most popular text-based agentic applications, which saw the bulk of initial development due to the language-only domain of LLMs. These applications leverage the rich semantic knowledge of LLMs to perform tasks ranging from information retrieval to complex software engineering. We will then turn to the most prominent multimodal application areas in which most research on agentic systems has been conducted in recent years.

| Application Domain | Evaluation Category | Primary Metric(s) | Representative Frameworks |
|---|---|---|---|
| **Robotics & Physical Embodiment** | Sim/Real-World Task Completion | Task success rate, planning success rate | SayCan, Inner Monologue, Code as Policies, PaLM-E, RT-2, OpenVLA, Octo, Magma |
| | Domain-Specific Driving Metrics | L2 distance, collision rate | EMMA |
| **GUI & Web Navigation** | Programmatic Functional Correctness | State validation, exact/substring match | WebAgent, AutoDroid, AssistGUI, AutoWebGLM, SeeAct |
| | Deterministic Benchmark Accuracy | Element accuracy, grounding accuracy, step success rate | CogAgent, SeeClick, GUI-Owl, Ferret-UI, Pix2Act, WebGUM |
| | Human Evaluation | Human preference, task completion | WebGPT, AppAgent, Mobile-Agent, MM-Navigator |
| | LLM-as-a-Judge | GPT-4V trajectory evaluation; GPT-4 fuzzy matching (partial) | WebVoyager; WebArena (info-seeking subset only) |
| **Long-Form Video Understanding** | Deterministic Automated Metrics | MCQ accuracy, temporal IoU, retrieval metrics | VideoAgent, LLoVi, DoraemonGPT, VideoMind, VLog |
| **Multimedia Content Generation & Editing** | Deterministic Benchmark Scores | T2I-CompBench, FID, CLIP Score, objective audio metrics | VISPROG, GenArtist, CoSTA*, FaSTA*, NExT-GPT |
| | Human Evaluation | MOS, human preference studies | AudioGPT, WavCraft, WavJourney, LAVE, CLOVA |

Table 1: This table summarizes the evaluation metrics used in various representative frameworks. It also shows how most frameworks have preferred deterministic metrics like *task success rate* and *grounding accuracy* over LLM-based judgments.

## 4.1 Text-only Applications

**Question Answering and Self-Correction** The earliest application of LLM-based agents focused on overcoming the static nature of pre-training data by connecting models to the internet. WebGPT (Rei-ichiro Nakano et al., 2021) pioneered this by training agents to use a text-based browser to issue search queries, scroll through results, and cite sources. Building on this, ReAct (Yao et al., 2022b) demonstrated that interleaving reasoning traces with environment actions significantly improves performance on multi-hop question-answering. Reflexion (Shinn et al., 2023a) introduced a *verbal reinforcement learning* mechanism which forces the agent to verbally reflect on its failures and store textual *lessons* in memory, rather than update it model weights. This self-correction loop allowed agents to achieve state-of-the-art performance on ALFWorld and HotpotQA by learning from experience within a text-only context.

**Multi-Agent Collaboration and Software Development** Text-based agents have also found success in software engineering, powered by foundational code-generation models such as Codex (Chen, 2021), which demonstrate that LLMs can solve competitive programming problems and serve as engines for coding agents. This capability evolved into multi-agent frameworks like MetaGPT (Hong et al., 2023), which mimics real-world software companies by assigning specialized roles such as Product Managers, Architects, and Engineers

to different LLM instances. By strictly defining standard operating procedures, these agents can collaboratively generate complex software repositories from high-level instructions. This paradigm of multi-agent collaboration was pioneered by Generative Agents (Park et al., 2023), which created a simulation of 25 agents with independent memories and reflections that engaged in complex, emergent social behaviors. This architecture demonstrated that *collaboration* can emerge from text-based agents interacting via natural language, a principle that features in frameworks such as ChatDev (Qian et al., 2023) for agentic software development.

**Generalist Tool and API Calling** Going beyond specific domain applications, there has been a focus on creating generalist agents capable of mastering massive libraries of digital tools. ToolFormer (Dwivedi-Yu et al., 2023) introduced a self-supervised method in which the LLM learns to invoke APIs by predicting calls as natural tokens. To scale this to massive toolsets where demonstrations exceed context windows, ToolkenGPT (Hao et al., 2023) represents tools not as text descriptions but as learnable token embeddings, which allows the LLM to call thousands of tools in an autoregressive manner. For real-world applications, RestGPT (Song et al., 2023) connects LLMs to complex RESTful APIs, such as movie databases or music players, using a coarse-to-fine online planning framework that handles reading of API documentation and parsing JSON responses. Finally, ToolLLM (Qin et al., 2023) truly generalizes this agentic capability by instruction-tuning models on 16,000+ real-world APIs.

> **Note:** Text-only applications use language as the primary interface for reasoning, planning, and tool orchestration. Even in digital environments, many systems first convert structured inputs such as web pages or GUI states into text-based representations, which makes them efficient and easy to control but also limits access to visual and temporal details that may matter for more complex tasks.

## 4.2 Robotics and Physical Embodiment

One of the essential aspects of building agents for embodied applications is *grounding*. Embodied agents must be grounded in their perceptual inputs to plan actions successfully and fulfill a task. While LLM/LMM-based agents have come up with various approaches to ground their sensory inputs, their outputs are usually in the form of language instructions or actions. To evaluate such frameworks, they must be tested in virtual text-based environments, such as ALFWorld (Shridhar et al., 2020) and WebShop (Yao et al., 2022a). While these approaches have been successful for multi-step reasoning and planning in complex tasks, there is a clear shift toward single-model agents trained end-to-end to generate physical actuation signals for accomplishing a particular task. Such agents can be evaluated on real-world datasets such as BridgeData V2 (Walke et al., 2023) and GNM (Shah et al., 2022). Table 2 summarizes the details of all the agents of this domain that have been surveyed in our paper.

**Grounded Planning with LLMs** The initial wave of embodied agents focused on bridging the gap between the semantic knowledge of Large Language Models (LLMs) and the physical capabilities of robots. SayCan (Ahn et al., 2022) pioneered this by acting as an orchestrator that evaluates the feasibility of actions based on two probabilities, *language probability* and *affordance probability*. This ensures that plans are not only semantically valid but also physically executable. Building on this, Inner Monologue (Huang et al., 2022) introduced a closed-loop feedback mechanism in which the agent processes environmental observations, such as *action success*, to dynamically replan during execution. Code as Policies (Liang et al., 2023) further evolved this paradigm by prompting LLMs to generate executable Python code rather than simple action lists. This enables agents to perform complex actions that previously required spatial-geometric reasoning and a precise value function. This effectively turning the LLM into a real-world policy generator that was grounded by perception.

> **Note:** Comparing these early frameworks reveals that bridging the semantic-to-physical gap requires moving beyond static affordance checks to dynamic, code-based policy generation that explicitly handles spatial constraints and environmental feedback.

**Multimodal Reasoning with MLLMs** As Multimodal Large Language Models (MLLMs) matured, frameworks began leveraging their native visual understanding for more sophisticated planning. ViLa ("Look Before You Leap") (Hu et al., 2023b) utilizes GPT-4V to integrate perceptual data directly into the reasoning

process, enabling a richer understanding of spatial information and object attributes that earlier text-only planners could not capture. Similarly, GPTArm (Zhang et al., 2025a) employs a hierarchical task-processing framework in which GPT-4V handles high-level intent understanding and error recovery, while specialized modules handle precise grasping. COME-robot (Zhi et al., 2025a) extends this by introducing a dual-loop system: a perception module for environment exploration and a closed-loop feedback mechanism for adaptive planning, achieving significantly higher success rates in open-ended tasks. Finally, EMMA (Hwang et al., 2024) demonstrates the adaptability of this approach in autonomous driving, treating motion planning as a multimodal reasoning task where raw camera inputs are mapped directly to trajectory predictions using a unified world model built on top of Gemini (Gemini Team, 2024).

**Note:** The evolution of these MLLM planners demonstrates that complex real-world control cannot rely on flat, single-stream reasoning; it requires hierarchical architectures where high-level semantic intent is decoupled from low-level, high-frequency manipulation.

**Vision-Language-Action Models** The current frontier of embodied agency lies in end-to-end models that fuse perception, reasoning, and control into a single architecture. PaLM-E (Danny Driess et al., 2023) advanced this by injecting multimodal sensor data directly into the language model's embedding space, enabling *embodied reasoning*, where plans are grounded in real-time physical states. Magma (Yang et al., 2025) further refines this with spatial and temporal awareness, using adapter-style training to process Set-of-Marks prompts for precise manipulation. RT-2 (Ichter et al., 2023) represents a shift toward foundational models that directly output tokenized robot actions from visual inputs, leveraging pre-training to generalize to novel objects. Octo (Octo Model Team et al., 2024) is a flexible, open-source generalist policy trained on the massive Open X-Embodiment dataset (O'Neill et al., 2024) and capable of adapting to different camera configurations and robot embodiments via a transformer-diffusion architecture. The VLA approach has been democratized by OpenVLA (Kim et al., 2024), which enables fine-tuning a small language model such as Llama-2 7B (Touvron et al., 2023) for specific robot action policies using LoRA (Hu et al., 2021).

More recently, the state-of-the-art has advanced with $\pi_0$ (Black et al., 2024) and its successor $\pi_{0.5}$ (Black et al., 2025), which introduce a family of vision-language-action flow models utilizing continuous flow-matching to directly generate smooth, high-frequency motor commands from visual inputs, scaling robustly to open-world environments. Moving beyond purely visual inputs, VTAM (Yuan et al., 2026) expands the sensory grounding of VLA models by integrating predictive tactile sensing into a generative world model, enabling foresight in complex contact-rich manipulation tasks. Extending these unified architectures to humanoid embodiments, GR00T N1 (GR00T Team et al., 2025) introduces a 2.2-billion-parameter open-source VLA with a dual-system design, where a vision-language backbone handles scene understanding, and a DiT-based flow-matching policy generates high-frequency motor commands, demonstrating strong language-conditioned bimanual manipulation on the Fourier GR-1 humanoid. A complementary line of work integrates explicit reasoning into the action-generation pipeline. CoT-VLA (Zhao et al., 2025) introduces visual chain-of-thought reasoning by generating subgoal images autoregressively before outputting physical actions, outperforming prior VLAs by 17% on real-world manipulation tasks. HALO (Shou et al., 2026) further unifies this paradigm through embodied multimodal chain-of-thought reasoning within a Mixture-of-Transformers architecture, addressing long-horizon and out-of-distribution scenarios by jointly performing textual task reasoning, visual foresight, and action prediction.

**Multi-Agent Systems** While the frameworks discussed above largely operate as single-agent systems, a complementary line of research explores how multiple specialized agents can collaborate to solve tasks that exceed the capacity of any individual model. RoCo (Mandi et al., 2024) exemplifies this by enabling groups of robots to coordinate through LLM-mediated dialogue, where agents negotiate task assignments and resolve spatial conflicts verbally before acting, demonstrating that natural language can serve as an effective coordination protocol even in physically grounded settings. Co-NavGPT (Yu et al., 2023) extends this to cooperative visual semantic navigation, where a shared Vision Language Model (VLM) planner distributes exploration responsibilities across a robot team based on real-time visual sub-maps. For complex environments, SMART-LLM (Kannan et al., 2024) decomposes high-level instructions and allocates sub-tasks to heterogeneous robots based on their specific skill sets and constraints, even dynamically forming multi-robot coalitions when a sub-task exceeds a single robot's capabilities. Integrating these multi-agent

coordination strategies with the unified perception-action tokenization of VLA models, such as RT-2 and OpenVLA, represents a promising and underexplored direction for scaling robotic agency to tasks requiring simultaneous collaboration across multiple embodiments.

> **Note:** Ultimately, the rapid democratization of VLA models highlights that the future of robotics lies in abandoning modular perception-reasoning pipelines in favor of unified, end-to-end architectures where visual inputs are mapped directly to physical actuation tokens, heavily relying on parameter-efficient fine-tuning to scale across diverse robot embodiments.

| Framework | Model Backbone | Tools/APIs | Training Strategy | Fusion Strategy |
|---|---|---|---|---|
| **Delegated Fusion (Tool-Based Perception)** | | | | |
| SayCan | PaLM | Learned policies for affordance | Zero-shot; skills are pre-trained via RL/BC | Combined probability of skill utility and execution success |
| Inner Monologue | InstructGPT, PaLM | Object Detectors, VQA models | Few-shot prompting | Processes textual feedback directly into LLM prompt |
| Code as Policies | Codex/GPT-3 | Perception APIs (ViLD, MDETR), Control Primitives | Few-shot prompting | Perception outputs in text are converted into code by the LLM |
| **Late Fusion (Modality-Specific Encoding)** | | | | |
| PaLM-E | PaLM | None (E2E model) | E2E training of encoders with frozen LLM | Input tokens are projected into the embedding space of LLM |
| RT-2 | PaLM-E / PaLI-X | None (E2E model) | Fine-tuning on VQA & trajectories | Robot actions tokenized with text/vision |
| OpenVLA | Llama 2 (7B) | None (E2E model) | LoRA Fine-tuning on Open X-Embodiment | Visual features are projected into the LLM embedding space |
| Magma | Llama 3 (8B) | None (E2E model) | Pre-training on multimodal data using SoM/ToM | ConvNeXt (img/vid) encodings are fed into the LLM embedding space |
| **Early Fusion (Unified Embedding Space) & API-based Frameworks** | | | | |
| Octo | Transformer (ViT) | None (E2E model) | Pre-trained on Open X-Embodiment | Block-wise attention mechanism between input and text tokens |
| ViLa | GPT-4V | None (primitive skills only) | Zero-shot prompting | Direct visual reasoning without intermediate affordance models |
| GPTArm | GPT-4V | YOLOv10 (for object detection) | Zero-shot prompting | GPT-4V processes visual observations directly |

| Framework | Model Backbone | Tools/APIs | Training Strategy | Fusion Strategy |
|---|---|---|---|---|
| COME-robot | GPT-4V | Perception APIs | Zero-shot prompting | GPT-4V processes visual observations directly |

Table 2: A summary of various representative frameworks under the **Robotics and Physical Embodiment** application. This table summarizes the content of Section 4.2 by classifying frameworks according to modality fusion and orchestration strategies. In classifying frameworks by fusion technique, we grouped early-fusion agents and API-based systems, as multimodal API-based frameworks are modular and can be replaced with different models.

### 4.3 GUI and Web Navigation

Agents operating in digital environments must be able to bridge the gap between high-level user intent and the pixel-level information in graphical user interfaces (GUIs). Unlike robotics, where physical affordances constrain actions, digital agents face the challenge of massive, dynamic action spaces in web pages and software applications, which contain thousands of digital elements that can be interacted with. Table 3 summarizes the details of all the agents of this domain that have been surveyed in our paper.

**Text-Based Tool-Augmented Agents** The first generation of digital agents relied on parsing tools to convert visual interfaces into text-based representations that standard LLMs could process. WebGPT (Reiichiro Nakano et al., 2021) demonstrated that LLMs could browse the web by interacting with a text-based environment, converting HTML into simplified text to issue search queries and scroll commands. Building on this, WebGLM (Liu et al., 2023a) introduced an efficient retrieval-augmented QA system that leverages an LLM-based retriever to process simplified HTML for accurate response generation. To handle long-context HTML documents, WebAgent (Gur et al., 2023) utilized HTML-T5 (Gür et al., 2023), a domain-specific encoder, to decompose instructions and synthesize Python programs for web automation. Addressing the specific constraints of mobile interfaces, AutoDroid (Wen et al., 2024) bridged the gap by converting mobile GUIs into a simplified HTML representation, enabling off-the-shelf LLMs to traverse application states without manual effort. Moving to desktop productivity, AssistGUI (Gao et al., 2023a) employed a sophisticated GUI parser that integrates OCR and icon detection to update a text-based planner and critic, thereby managing complex software such as Adobe Premiere. Most recently, AutoWebGLM (Lai et al., 2024) advanced web navigation by designing an HTML simplification algorithm and employing reinforcement learning with rejection sampling to bootstrap the agent's comprehension of text-dense web pages.

**Note:** Ultimately, the evolution of text-augmented agents exposes a hard limitation: attempting to navigate GUIs via HTML-to-text conversion is fundamentally bottlenecked by context limits, and aggressive text simplification inherently strips away the crucial spatial and layout information required for robust navigation.

**Fine-tuned Late-fusion Multimodal Agents** To overcome the information loss inherent in text-only conversions, researchers developed specialized multimodal models trained explicitly for UI understanding. These works typically employ a late-fusion architecture, where a vision encoder projects screen features into an LLM's embedding space via an adapter. WebGUM (Furuta et al., 2023) combines a ViT encoder with a T5 (Raffel et al., 2023) language model, fine-tuned on webpage screenshots and HTML history. Pix2Act (Shaw et al., 2023) builds on the Pix2Struct (Lee et al., 2023) architecture to output valid actions directly from pixels without HTML access. For high-resolution perception, CogAgent (Hong et al., 2024) introduces a specialized visual encoder that processes $1120 \times 1120$ images to detect small icons. Ferret-UI (You et al., 2024) employs an adaptive gridding strategy to handle variable mobile aspect ratios. To address the grounding bottleneck, SeeClick (Cheng et al., 2024) fine-tunes the Qwen2-VL (Bai et al., 2024) architecture on a large-scale dataset of GUI screenshots and action pairs, enabling it to predict precise screen coordinates for elements that generalist models miss. Similarly, GUI-Owl (Ye et al., 2025) introduces an end-to-end foundational model built on Qwen2-VL, enabling robust multi-turn interaction across both mobile and desktop environments by treating GUI navigation as a specialized vision-language task rather than a general prompting problem.

> **Note:** Comparing these late-fusion models reveals that general-purpose vision encoders are insufficient for digital navigation; achieving state-of-the-art performance requires architectures specifically optimized for extreme high-resolution processing and explicit, pixel-level coordinate grounding.

**Natively Multimodal Agents** The current state-of-the-art uses general-purpose, natively multimodal Large Multimodal Models (LMMs) such as GPT-4V or Gemini as the reasoning core. These agents do not require model training; instead, they focus on agentic frameworks, prompting strategies, memory modules, and tool-use to ground the LMM's capabilities. SeeAct (Zheng et al., 2024) leverages GPT-4V (OpenAI, 2023) to act on live websites, using a "visual grounding" strategy to iteratively refine plans based on visual feedback. WebVoyager (He et al., 2024a) extends this to end-to-end web navigation, parsing screenshots directly to identify interactable elements. In the mobile domain, AppAgent (Zhang et al., 2023a) enables LMMs to operate smartphone apps by mimicking human taps and swipes solely from visual cues. Mobile-Agent (Wang et al., 2024c) enhances this by adding a "self-reflection" module, allowing the LMM to correct its own navigation errors if a visual state change doesn't match its expectation. One fundamental ability in GUI agents is grounding. ShowUI (Lin et al., 2025) trains a lightweight 2B vision-language-action model unifying perception, reasoning, and action, using UI-guided token selection to prune redundant visual tokens and improve zero-shot screenshot grounding. FocusUI (Ouyang et al., 2026) extends this efficiency direction with a position-preserving token selection strategy that compresses each run of dropped tokens into a single marker, retaining the positional continuity critical for grounding even under aggressive pruning. ShowUI-$\pi$ (Hu et al., 2025) instead targets a shared limitation: discrete click coordinates cannot express continuous trajectories such as dragging a progress bar. It reformulates GUI control as flow-based generation over cursor waypoints, unifying clicks and drags in one model and enabling real-time trajectory adjustment. Finally, MM-Navigator (Yan et al., 2023) comprehensively evaluates GPT-4V's agentic capabilities on iOS and Android screens. It demonstrates that, while multimodal models exhibit strong semantic understanding, they require auxiliary prompting strategies, such as Set-of-Marks or numbered tags, to achieve the precise coordinate grounding necessary for reliable interaction. Addressing these limitations, UI-TARS-2 (Wang et al., 2025a) unifies perception, reasoning, and memory through a stabilized multi-turn reinforcement learning framework, achieving state-of-the-art autonomous interaction across hybrid desktop, mobile, and terminal environments. Concurrently, Agent S2 (Agashe et al., 2025) tackles precise GUI localization through a Mixture-of-Grounding approach, delegating cognitive responsibilities across generalist planners and specialized visual execution modules to enhance long-horizon reliability.

Complementing these approaches, OpenCUA (Wang et al., 2025b) provides the first comprehensive open-source framework for computer-use agents, encompassing a cross-OS dataset of 22,600 tasks and end-to-end agent models, with OpenCUA-72B achieving 45.0% on OSWorld-Verified to establish a new state-of-the-art among open-source models. Addressing the persistent grounding bottleneck from a training perspective, InReAct (Wang et al., 2025c) proposes an inspire-then-reinforce framework that applies GRPO-based reinforcement fine-tuning to improve the precise localization of small GUI elements that generalist models frequently miss. Concurrently, GUI-Genesis (Yu et al., 2026a) tackles the efficiency bottleneck by synthesizing lightweight environments with verifiable code-native rewards for GUI agent post-training, reducing environment latency by $10\times$ and training costs by over \$28,000 per epoch compared to training on real applications.

**Multi-Agent Systems** Beyond standard monolithic frameworks, a growing body of work decomposes GUI and web navigation into modular or collaborative pipelines where distinct components handle specific cognitive responsibilities. While Cradle (Tan et al., 2024) exemplifies this within a single-agent architecture by utilizing six distinct modules (including information gathering, self-reflection, and action planning) to generalize across diverse desktop environments, other systems extend this decomposition into true multi-agent collaboration. Mobile-Agent-v2 (Wang et al., 2024b), already discussed above, separates planning from decision-making into distinct agent roles, addressing the context degradation problem that plagues single-agent architectures on long-horizon tasks. OpenAgents (Xie et al., 2023) similarly instantiates a multi-agent platform comprising specialized Data, Plugin, and Web agents, allowing complex user queries to be routed across agents with complementary competencies. From a data perspective, AgentTrek (Xu et al., 2024) demonstrates how multi-agent collaboration can address the data scarcity bottleneck inherent in GUI

agent training: it employs a VLM agent to execute tasks derived from web tutorials, while a separate VLM evaluator assesses trajectory correctness to synthesize large-scale, high-quality interaction data automatically.

**Note:** The reliance on these frameworks dictates a clear design lesson: while natively multimodal models eliminate the need for domain-specific training, their inherent weakness in spatial grounding requires developers to build robust auxiliary pipelines, specifically visual tagging overlays and explicit self-reflection loops to achieve reliable GUI interaction.

| Framework | Model Backbone | Tools/APIs | Training Strategy | Fusion Strategy |
|---|---|---|---|---|
| **Delegated Fusion (Tool-Based Perception)** | | | | |
| WebGPT | GPT-3 | Bing Web Search API | Behavior Cloning (BC) and Reward Modeling (RM) | Converts web pages into simplified text summaries |
| WebGLM | GLM-10B | Google Search API, HTML2Text, Contriever | SFT on bootstrapped QA, RLHF | LLM-augmented retriever distills web content into text references |
| WebAgent | HTML-T5, Flan-U-PaLM | Selenium WebDriver | Pre-training on HTML web data, fine-tuning on self-experience | Local & global attention to process long HTML structures, built on T5 backbone |
| AutoDroid | GPT-3.5/4, Vicuna-7B | Android UI Automator (XML) | Exploration-based memory injection, Fine-tuning (for local models) | Parses UI XML into simplified HTML-style text representation |
| AssistGUI | GPT-4 | PyWinAuto, Google OCR, YOLOv8 | In-context learning using Actor-Critic framework | Aggregates outputs from multiple vision tools into text descriptions |
| **Late Fusion (Modality-Specific Encoding)** | | | | |
| CogAgent | CogVLM-17B | None (E2E model) | Pre-training on text recognition and image captioning | Cross-attention fusion of image features with VLM decoder for high resolution understanding |
| SeeClick | Qwen-VL | None (E2E model) | GUI grounded pre-training, LoRA fine-tuning | Directly predicts coordinates without parsing HTML |
| GUI-Owl | Qwen2.5-VL | Virtual Env, PyAutoGUI | Pre-training, Scalable RL (TRPO/GRPO) | Unifies perception and action in a single policy network |
| Ferret-UI | Ferret (CLIP-ViT+Vicuna) | Apple Vision, Screen Recognition | SFT on UI tasks of increasing granularity | Encodes images and visual grids alongside text embeddings |

| Framework | Model Backbone | Tools/APIs | Training Strategy | Fusion Strategy |
|---|---|---|---|---|
| Pix2Act | Pix2Struct | Selenium | Pre-trained on a screenshot parsing task | Encodes images and visual grids alongside text embeddings |
| **Early Fusion (Unified Embedding Space) & API-based Frameworks** | | | | |
| AppAgent | GPT-4V | Android Debug Bridge, XML Parser | UI exploration and in-context learning | Processes interleaved image and text inputs directly |
| Mobile-Agent | GPT-4V | OCR tools, Grounding DINO, CLIP | Zero-shot prompting | GPT-4V handles perception; OCR/Det. handles localization |
| MM-Navigator | GPT-4V | OCR, IconNet, SAM | In-context learning through summary of history | Uses SoM tags for grounding GPT-4V's visual reasoning |
| WebVoyager | GPT-4V | Selenium, GPT-4V-ACT | In-context learning similar to ReAct | Overlays BBoxes and SoM on screenshots |
| SeeAct | GPT-4V | Playwright, DeBERTa | In-context learning | Standard/SoM prompting depending on grounding strategy |

Table 3: A summary of various representative frameworks under the **GUI and Web Navigation** application. This table summarizes the content of Section 4.3 by classifying frameworks according to modality fusion and orchestration strategies. In classifying frameworks by fusion technique, we grouped early-fusion agents and API-based systems, as multimodal API-based frameworks are modular and can be replaced with different models.

## 4.4 Multimedia Content Generation and Editing

In the domain of creative production, multimodal agents function as intelligent orchestrators that bridge the gap between high-level user intent and the complex parameters of generative foundation models. Unlike standard generation tasks, where a single prompt yields a single output, agentic frameworks in this space are responsible for multi-step planning, tool composition, and iterative refinement to produce coherent multimedia content. Agentic frameworks have been particularly successful in editing operations, as they can perform focused manipulations without distorting the remaining information in the operating environment. We have primarily observed a pattern similar to that in past application domains, as described below. Table 4 summarizes the details of all the agents of this domain that have been surveyed in our paper.

**Text-Based Tool-Augmented Agents** Early multimodal agents frequently circumvented the inability of standard LLMs to perceive non-textual data by converting inputs into intermediate textual representations or executable code. VISPROG (Tanmay Gupta, 2022) introduced a neuro-symbolic system that leverages the in-context learning capabilities of GPT-3 (Brown et al., 2020) to invoke predefined modules that incorporate various vision foundation models and traditional computer vision (OpenCV) library functions. These modules were used to perform various subtasks derived from the high-level instructions received by the LLM. In the audio domain, AudioGPT (Huang et al., 2023a) connects LLMs with audio foundation models by employing a *modality transformation* interface that converts spoken dialogue and audio into text via ASR and captioning, enabling the LLM to process user intent and coordinate tool use. Building on this modular approach, WavJourney (Liu et al., 2023b) uses LLMs to generate structured audio scripts that represent the spatiotemporal relationships among sound elements, which are then compiled into computer programs to invoke the generation models. Similarly, WavCraft (Liang et al., 2024) describes raw audio materials

in natural language to prompt an LLM, which then decomposes instructions into tasks for specific audio modules, facilitating content creation through code generation. Addressing the complexity of video editing, LAVE (Wang et al., 2024a) employs an LLM-powered agent to assist with ideation and operational tasks. To overcome the model's text-only limitation, LAVE automatically generates textual descriptions of video footage, which serve as the semantic foundation for the agent to plan and execute editing actions.

**Note:** The shared architecture of these early frameworks demonstrates that successfully orchestrating multimedia generation with text-only LLMs relies heavily on a neuro-symbolic approach, where the LLM is strictly confined to semantic routing and code generation, entirely offloading the actual media manipulation to specialized, external foundation models.

**Natively Multimodal Agents** Recent advancements have shifted towards natively multimodal agents and tuning-free frameworks that leverage pre-trained models without additional parameter updates. GenArtist (Wang et al., 2024f) utilizes a Multimodal LLM (GPT-4V) as a unified agent capable of directly perceiving images to plan and verify generation tasks. This agent constructs a planning tree to decompose complex prompts and executes tools while performing self-correction based on visual feedback. Focusing on cost-efficiency, CoSTA* (Gupta et al., 2025b) proposes a training-free agent that combines LLMs with A* search to find optimal toolpaths for image editing. It relies on a *Model Description Table* and benchmark data to navigate a graph of off-the-shelf tools, optimizing for quality and cost without fine-tuning the underlying models. Building on this, FaSTA* (Gupta et al., 2025a) introduces a *fast-slow planning* mechanism that mines reusable symbolic subroutines from successful toolpaths, significantly reducing search costs for recurrent editing tasks. In the realm of video editing, FLATTEN (Cong et al., 2023) enforces visual consistency through a tuning-free flow-guided attention mechanism, which integrates optical flow into the attention modules of pre-trained diffusion models. Similarly, UniEdit (Bai et al., 2025a) presents a unified, tuning-free framework that employs an inversion-then-generation pipeline to enable motion and appearance editing using pre-trained text-to-video generators. Finally, AnyV2V (Ku et al., 2024) offers a tuning-free framework that plugs into various image-to-video models to perform editing tasks by manipulating spatial and temporal features during the diffusion process.

**Note:** The evolution of these systems highlights that while native multimodal perception eliminates the text-conversion bottleneck, achieving efficient, temporally consistent editing requires pairing these agents with tuning-free algorithmic constraints (like A* search and optical flow guidance) to mitigate the massive computational costs of multi-step generation.

**Multi-Agent Systems** The creative domain has also seen the emergence of multi-agent frameworks that decompose content generation and editing into role-specialized collaborative loops, mirroring the division of labor observed in human creative workflows. CREA (Venkatesh et al., 2025), discussed above, is the most direct instantiation of this principle, assigning distinct roles such as Creative Director, Art Critic, and Refinement Strategist to separate agents that engage in iterative critique and revision cycles, yielding outputs that surpass single-agent baselines in both semantic alignment and creative diversity. ReelWave (Wang et al., 2025d) adopts a similar paradigm in the audiovisual domain, utilizing a role-playing production team to generate and synchronize multi-scene movie sound collaboratively. EditDuet (Sandoval-Castaneda et al., 2025) further explores this collaborative dynamic in video non-linear editing (NLE) through a two-agent architecture: an Editor agent uses software tools to search and sequence raw video clips into a draft timeline, while a Critic agent evaluates the assembly against the user's high-level request. Paper2Poster (Pang et al., 2026) generates academic posters from papers through a pipeline of specialized agents that handle content parsing, layout planning, and visual refinement under a coherent design objective. Paper2Video (Zhu et al., 2025) extends this to presentation videos, coordinating role-specialized agents across the aligned channels of slide generation, cursor grounding, subtitling, speech synthesis, and talking-head rendering to produce a complete academic video. This creates a closed feedback loop that refines the video sequence and reduces error accumulation across multi-step editing tasks.

| Framework | Model Backbone | Tools/APIs | Training Strategy | Fusion Strategy |
|---|---|---|---|---|
| **Delegated Fusion (Tool-Based Perception)** | | | | |
| VISPROG | GPT-3 | CLIP, ViLT, SD3, OpenCV functions | Few-shot learning using in-context editing examples | Off-the-shelf vision modules process input images |
| AudioGPT | GPT-3 | Whisper, DiffSinger, AudioLDM etc. | Zero-shot prompting | Uses task-specific tools to handle editing operations |
| WavCraft | GPT-4 | MusicGen, AudioSep, AudioSR | Uses in-context learning to generate code | LLM reasons over audio text descriptions and user queries to generate code |
| WavJourney | GPT-4 | AudioLDM, MusicGen, Bark, etc. | In-context learning to generate audio script | LLM reasons over audio text descriptions and user queries to generate audio script |
| LAVE | GPT-4 | LLaVA, Vector Store, ffmpeg | In-context learning to generate editing actions | LLM processes text summaries of videos from LLaVA to plan edits |
| CLOVA | GPT-4 | OWL-ViT, BLIP, CLIP, SD3 | Closed-loop learning by updating correct reasoning traces | LLM uses reflection to critique its performance through intermediate and final results |
| Audio-Agent | GPT-4 (TTA), Gemma-2B (VTA) | Auffusion | Fine-tuning required for Gemma to adapt to visual tokens | Video is converted into semantic tokens that align with the audio |
| **Early Fusion (Unified Embedding Space) & API-based Frameworks** | | | | |
| GenArtist | GPT-4V | SDXL, ControlNet, Grounding DINO, SAM | Few-shot prompting to guide MLLM planning | MLLM takes images natively along with auxiliary bboxes |
| CoSTA* | GPT-4o | Visual Perception/Editing Tools | Training-free; Utilizes A* search on a tool graph | LLM does hierarchical planning based on image and text inputs |
| FaSTA* | GPT-4o | Visual Perception/Editing Tools | Learns by mining successful tool executions | Natively uses image and text for fast-slow planning |

Table 4: A summary of various representative frameworks under the **Multimedia Content Generation and Editing** application. This table summarizes the content of Section 4.4 by classifying frameworks according to modality fusion and orchestration strategies. In classifying frameworks by fusion technique, we grouped early-fusion agents and API-based systems, as multimodal API-based frameworks are modular and can be replaced with different models.

### 4.5 Long Form Video Understanding and Retrieval

Processing long-form video introduces unique computational and reasoning challenges that distinctively separate it from static image analysis. A standard video generates hundreds to thousands of visual tokens per minute, resulting in a quadratic increase in attention costs that overwhelms standard context windows. While architectural solutions such as LongVU (Shen et al., 2024) and Video-XL (Shu et al., 2024) address this by expanding context and compressing tokens, agentic frameworks offer a different paradigm: treating video not as a single input to be processed, but as a dynamic environment to be explored and queried. Table 5 summarizes the details of all the agents of this domain that have been surveyed in our paper.

**Iterative Retrieval Agents** This category of agents addresses the token bottleneck by selectively retrieving only the frames relevant to a specific query, rather than encoding the entire video. AssistGPT (Gao et al., 2023b), whose plan–execute–inspect–learn loop interleaves reasoning with tool invocation, inspecting intermediate outputs to decide what to query next rather than processing all inputs in a single pass. VideoAgent (Wang et al., 2024e) exemplifies this approach by using an LLM (GPT-4) as a central orchestrator that iteratively generates natural language queries to search a database of video frames encoded by CLIP (Radford et al., 2021b). By engaging in multi-turn dialogue with the video storage, querying, retrieving, and refining, it achieves state-of-the-art performance on benchmarks such as EgoSchema (Zhuang et al., 2023) while processing 20x fewer frames than dense sampling methods. Similarly, LLoVi (Zhang et al., 2024) employs a two-stage decomposition strategy where a vision-captioner first translates densely sampled frames into textual descriptions. An LLM then reasons over this *video-language* representation, thereby performing temporal reasoning in text space without the computational overhead of visual encoders.

> **Note:** Comparing these approaches reveals a shared foundational limitation: overcoming the video token bottleneck in early frameworks required heavily lossy proxies, either relying on external vector searches or flattening the video into text descriptions, which inherently sacrifices fine-grained, continuous visual details for computational feasibility.

**Specialized Fine-tuned Reasoning Agents** For more complex tasks requiring deep temporal grounding, agents utilize specialized roles or adapters to manage different aspects of reasoning. VideoMind (Liu et al., 2025) introduces a "Chain-of-LoRA" architecture, where a single base MLLM (Qwen2-VL) dynamically switches between lightweight LoRA adapters trained for specific agentic roles: a Planner to decompose questions, a Grounder to find relevant timestamps, and a Verifier to check evidence. This allows a relatively small model to outperform significantly larger ones by activating only the necessary reasoning circuits for each step of the video analysis. VLog (Lin & Shou, 2025) takes a different approach by redefining the representation of video itself; it treats video narrations as compositional vocabulary units, enabling the agent to perform "generative retrieval" where finding a specific event in an hour-long video becomes as fast as predicting the next word in a sentence.

> **Note:** The success of these specialized agents highlights a critical architectural shift: achieving deep temporal grounding without massive compute scaling requires modularizing the reasoning process itself either by dynamically swapping task-specific model weights on the fly (via LoRA) or fundamentally restructuring how video data is represented.

**Natively Multimodal Agents** The shift towards native MLLMs is also reshaping video agents. Models like Gemini 1.5 Pro (Gemini Team, 2024) and GPT-4o (OpenAI, 2024) can now ingest hour-scale videos directly into their context windows, allowing for end-to-end reasoning without external retrieval loops. However, benchmarks like Video-MME (Fu et al., 2024) and EgoSchema (Zhuang et al., 2023) reveal that while these models excel at short-term recognition, they still struggle with causal reasoning over long horizons (e.g., "Why did the character take the key in the first scene?"). To bridge this gap, frameworks such as DoraemonGPT (Huang et al., 2025) wrap native MLLMs in agentic loops that explicitly model dynamic scenes and track object states over time, demonstrating that even with massive context windows, agentic structure is often necessary for robust long-form understanding. Advancing this retrieval paradigm, VideoRAG (Ren et al., 2025) implements a dual-channel architecture that combines graph-based textual knowledge grounding with multi-modal context encoding, allowing the agent to accurately process unlimited-length videos across

external databases. Similarly, VideoDeepResearch (Yuan et al., 2025) surpasses extended-context MLLMs by employing a text-only large reasoning model that selectively queries video content through an orchestrated suite of multi-modal retrieval and perception tools. DVD (Deep Video Discovery) (Zhang et al., 2025c) advances this agentic paradigm by replacing rigid predefined workflows with autonomous adaptive planning. Equipped with search-centric tools operating on a multi-granular video database, the agent dynamically selects tools and iteratively refines its reasoning based on gathered information, achieving 74.2% accuracy on the challenging LVBench dataset and substantially surpassing all prior works.

> **Note:** Ultimately, the reliance on these external frameworks proves a crucial point for the field: an expanded context window solves the input bottleneck, but it does not solve the reasoning bottleneck; achieving true long-form causal understanding still mandates external agentic structures to explicitly track states and causality across time.

| Framework | Model Backbone | Tools/APIs | Training Strategy | Fusion Strategy |
|---|---|---|---|---|
| **Delegated Fusion (Tool-Based Perception)** | | | | |
| VideoAgent | GPT-4 | VLM: LaViLa, CogAgent; Retrieval: CLIP | Iterative sampling of frames based on LLM reflection | LLM processes video frames as text caption from the VLM |
| LLoVi | GPT-3.5/4 | LaViLa, BLIP-2, LLaVA, EgoVLP | Training free; Prompt-based summarization of short clip captions | LLM process the caption for short clips and summary for reasoning |
| DoraemonGPT | GPT-3.5 | YOLOv8, BLIP-2, Whisper, Google Search | Training free; Uses in-context learning with a MCTS planner | LLM queries from text-based symbolic memory for retrieving specific information |
| **Late Fusion (Modality-Specific Encoding)** | | | | |
| VLog | GPT-2, Qwen2-7B | LLaVA, Qwen2.5 used for vocab upgrade | LLM is fine-tuned to map inputs into narration | Visual embeddings from SigLIP are appended to text query embeddings |
| VideoMind | Qwen2-VL | LoRA-adapted multi-agents | Chain-of-LoRA fine-tuning with adaptors trained for specific roles | The base MLLM uses late-fusion to process multimodal tokens |

Table 5: A summary of various representative frameworks under the **Long Form Video Understanding and Retrieval** application. This table summarizes the content of Section 4.5 by classifying frameworks according to modality fusion and orchestration strategies. In classifying frameworks by fusion technique, we grouped early-fusion agents and API-based systems, as multimodal API-based frameworks are modular and can be replaced with different models.

## 5 Performance Analysis

The architectural progression from delegated perception through late-fusion to early-fusion approaches yields measurable performance gains across application domains. This section examines the performance implications of these architectural progressions across the four primary application domains established in Section 4: robotics and physical embodiment, GUI and web navigation, long-form video understanding, and multimedia

content generation. We correlate specific architectural innovations with quantitative improvements, demonstrating how tighter integration between perception, reasoning, and action addresses information bottlenecks inherent in earlier designs.

## 5.1 Robotics and Physical Embodiment

Delegated perception approaches establish a baseline level of competence but suffer from information loss during text conversion. SayCan (Ahn et al., 2022) achieves 84% planning success and 74% execution success by grounding LLM outputs through affordance scoring, where the model evaluates both semantic validity and physical feasibility of proposed actions. While this dual-probability framework enables reliable execution of known skills, the architecture remains constrained to predefined skill primitives and cannot generalize to novel objects or instructions outside its training distribution. The transition to late-fusion architectures begins addressing these limitations. PaLM-E (Danny Driess et al., 2023) injects continuous sensor embeddings directly into a 562-billion parameter language model, enabling reasoning over both linguistic instructions and real-time sensory data. This architectural choice preserves geometric and spatial information that text conversion discards, supporting more sophisticated embodied reasoning.

Early-fusion vision-language-action models yield the most substantial improvements by unifying perception and action within a single representational space. RT-2 (Ichter et al., 2023) tokenizes robot actions alongside vision-language tokens, enabling semantic transfer from web-scale pretraining directly to motor control. This design achieves 62% success on novel object tasks, compared with RT-1's 32%, a 30 percentage-point gain attributable to the model's ability to leverage internet-scale knowledge to interpret unseen commands and reason about object properties. The architecture achieves 90% success on the Language Table simulation while demonstrating emergent capabilities, such as following commands involving object relationships that were never observed during robotic training. OpenVLA (Kim et al., 2024) demonstrates that this early-fusion approach scales efficiently with modest parameters. Trained on 970,000 robot demonstrations, the 7-billion-parameter model outperforms the 55-billion-parameter RT-2-X by 16.5% in absolute success across 29 evaluation tasks and surpasses Diffusion Policy by 20.4%. These results suggest that architectural integration matters more than parameter count for embodied generalization. Octo (Octo Model Team et al., 2024) extends this paradigm to cross-embodiment transfer, training on 800,000 trajectories across multiple robot platforms and demonstrating that unified vision-language-action architectures can generalize across different camera configurations and physical embodiments. Table 6 below shows a more intuitive comparison of frameworks in this domain. Based on the results, we see that OpenVLA has significantly outperformed RT-2-X (55B) on major evaluation benchmarks, suggesting that our conclusion about the efficiency of late-fusion architectures is correct. The $\pi_0$ family of models further advances this trajectory by replacing discrete action tokenization with continuous flow matching. $\pi_0$ (Black et al., 2024) achieves 50 Hz control frequency from a 3B-parameter VLM backbone, enabling dexterous tasks such as laundry folding that require temporally smooth motor commands beyond the resolution of discrete token vocabularies. $\pi_{0.5}$ (Black et al., 2025) extends this to open-world generalization through heterogeneous co-training, performing 10 to 15 minute manipulation sequences in previously unseen homes.

| Benchmarks | Octo (93M) 
 *reported by OpenVLA* | Octo (93M) 
 *reported by Octo* | RT-2-X (55B) | OpenVLA (7B) | $\pi_{0.5}$ (3B) |
|---|---|---|---|---|---|
| Google Robot | 26.7% | ~80% | 78.3% | **85.0%** | - |
| BridgeData V2 WidowX | 20% | ~50% | 50.6% | **70.6%** | - |
| LIBERO Simulation | 75.1% | - | - | 76.5% | **96.8%** |

Table 6: **Framework Comparison.** This table shows the side-by-side performance comparison of representative agentic frameworks across common benchmarks in the robotics domain. It is important to note that the benchmark scores of Octo vary significantly, depending on the source, as mentioned in the table.

**Note:** In embodied settings, *late-fusion* VLA models like **OpenVLA** demonstrate that architectural integration matters more than raw parameter count: a 7B model outperforms 55B alternatives by 16.5% through unified perception-action tokenization rather than scale alone.

## 5.2 GUI and Web Navigation

Text-based approaches establish functional baselines through language-mediated interaction with digital interfaces. WebGPT (Reiichiro Nakano et al., 2021) achieves 56% preference over human demonstrators and 69% preference over the highest-voted Reddit answers on ELI5 question-answering by converting HTML to simplified text representations. However, this conversion discards visual layout information that is critical for interpreting ambiguous interfaces, where element positioning conveys semantic meaning. Late-fusion architectures address this limitation through dedicated visual encoders that preserve spatial information. CogAgent (Hong et al., 2024) processes 1120×1120 resolution images via a specialized high-resolution encoder, enabling detection of small icons and fine-grained UI elements invisible to text-only parsing or lower-resolution visual models. This architectural choice directly addresses the information bottleneck in which accessibility trees and HTML parsing fail to capture visual distinctions among similar elements.

Native multimodal agents leverage general-purpose vision-language models for end-to-end visual reasoning. SeeAct (Zheng et al., 2024) and WebVoyager (He et al., 2024a) process screenshots directly using GPT-4V, enabling both perception and action generation within a unified framework. Yet benchmark evaluations reveal persistent performance gaps that highlight remaining architectural challenges. On WebArena (Zhou et al., 2024b), GPT-4-based agents achieve a 14.41% success rate compared to 78.24% human performance. VisualWebArena (Koh et al., 2024) exposes additional limitations: GPT-4V with a Set-of-Marks representation achieves 16.4% success, compared with 88.7% human performance, and struggles particularly on optical character recognition tasks, where text-heavy interfaces require precise visual parsing. This persistent 60-70 percentage point gap indicates that current architectures, while superior to text-only approaches, still lack the precise coordinate grounding, multi-step planning, and error recovery mechanisms required for reliable GUI interaction. Table 7 below shows a more intuitive comparison of the representative frameworks in this domain across common GUI and Web navigation benchmarks. We have observed that the frameworks are mostly specialized based on the application area. For web navigation, Text/HTML-based architectures such as AutoWebGLM and WebAgent perform better because they can leverage the underlying web code for high accuracy. However, GUI navigation is much more complicated as the underlying code for these interfaces cannot be easily extracted, and the graphical intricacies require visual perception. Recent advances are beginning to narrow this gap. UI-TARS-2 (Wang et al., 2025a) demonstrates that multi-turn RL with a data flywheel can push a native GUI agent to 47.5% on OSWorld and 88.2% on Online-Mind2Web. Agent S2 (Agashe et al., 2025) achieves comparable gains through architectural decomposition: by delegating grounding to specialized visual modules and planning to generalist reasoners, it reaches 34.5% on the OSWorld 50-step evaluation, a 32.7% relative improvement over monolithic baselines. The convergence of these orthogonal approaches suggests that the performance gap may be more tractable through targeted architectural innovations than through further scaling of generalist models alone.

| Benchmark | GPT-4 Baseline | WebAgent | AutoWebGLM | CogAgent | SeeClick |
|---|---|---|---|---|---|
| Mind2Web | 30.9% | 46.7% | 59.5% | **58.2%** | **20.9%** |
| MiniWob++ | 32.1% | 85.6% | 89.3% | - | **67.0%** |
| AITW | 50.54% | - | - | **76.88%** | **66.4%** |
| ScreenSpot | 16.2% | - | - | **47.4%** | **53.4%** |

Table 7: **Framework Comparison.** This table shows the side-by-side performance comparison of representative agentic frameworks across common benchmarks in the GUI and web navigation domain.

**Note:** Despite architectural advances from delegated perception to native multimodal processing, GUI agents exhibit a persistent 60-70 percentage point gap versus human performance on benchmarks like **WebArena** and **VisualWebArena**, indicating that precise coordinate grounding, multi-step planning, and error recovery remain fundamentally unsolved challenges.

### 5.3 Long-Form Video Understanding

The computational burden of processing hour-scale video creates unique architectural pressures distinct from those of image or short-clip understanding. Standard attention mechanisms exhibit quadratic complexity, which overwhelms context windows when processing thousands of frames. Iterative retrieval agents address this through query-driven frame selection rather than dense encoding, treating video as an environment to be explored rather than a single input to be processed. LLoVi (Zhang et al., 2024) exemplifies this approach by converting sampled frames into textual descriptions via a vision-language captioner, and then performing temporal reasoning entirely in the language space. This two-stage decomposition achieves 52.2% accuracy on EgoSchema, an 18.1 percentage point improvement over prior methods that attempted full video encoding. By shifting reasoning from visual to linguistic representations, the architecture sidesteps the computational bottleneck while preserving semantic content sufficient for question-answering tasks.

VideoAgent (Wang et al., 2024e) refines this paradigm by introducing CLIP-based retrieval within an LLM orchestration loop. The agent iteratively queries a frame database, retrieves relevant segments, and refines its understanding through multi-turn interaction. This selective attention mechanism achieves 54.1% on EgoSchema while processing only 8.4 frames on average, yielding 2-30× efficiency gains over dense sampling approaches. The 3.8 percentage-point improvement in accuracy over LLoVi demonstrates that strategic content selection not only reduces computational cost but also aligns with the inherent sparsity of task-relevant information in long-form video. VideoMind (Liu et al., 2025) introduces architectural specialization through role-specific LoRA adapters for planning, temporal grounding, and verification. This Chain-of-LoRA (Wenhan Xia, 2024) design enables a 2B parameter model to outperform 78B parameter alternatives (a 39× size difference) by activating only the reasoning circuits required for each analysis step. The architecture demonstrates that targeted specialization can substitute for parameter scale when task decomposition aligns with modular adapter design. VLog (Lin & Shou, 2025) proposes a fundamentally different approach through generative retrieval, treating video narrations as compositional vocabulary units rather than generating full descriptions. This reformulation reduces temporal search complexity from linear to logarithmic, achieving 10-20× speedup over generative baselines. A 124M-parameter model matches the performance of LLaMA-2-7B (56× larger), demonstrating that algorithmic innovation in retrieval architecture can obviate the need for massive parameter scaling when the representational structure aligns with the task's information-retrieval demands. Table 8 below shows a more intuitive comparison of frameworks in this domain. The results show that VideoAgent and LLoVi achieve the highest accuracy on standard video QA benchmarks. LLoVi achieves this through a multi-round summarization of video clips while VideoAgent uses an LLM to iteratively select frames. Retrieval-augmented and tool-using approaches extend these gains further. VideoRAG (Ren et al., 2025) processes videos of unlimited length through a dual-channel architecture combining graph-based knowledge grounding with multi-modal retrieval, enabling cross-video reasoning over corpora spanning hundreds of hours. VideoDeepResearch (Yuan et al., 2025) demonstrates that a text-only reasoning model paired with modular multi-modal tools surpasses dedicated MLLMs on long-video benchmarks, achieving 9.6% improvement over prior state-of-the-art on MLVU while outperforming GPT-4o and Gemini 1.5 Pro, suggesting that the bottleneck in long-form video understanding lies more in reasoning and retrieval strategy than in visual encoding capacity.

| Benchmark | DoraemonGPT | VLog | LLoVi | VideoAgent |
|---|---|---|---|---|
| EgoSchema | - | 43.1% | **52.2%** | **54.1%** |
| NExT-QA | 55.7% | - | **73.8%** | **71.3%** |

Table 8: **Framework Comparison.** This table shows the side-by-side performance comparison of representative agentic frameworks across common benchmarks in the long-form video domain.

**Note:** Selective retrieval and algorithmic innovation can substitute for parameter scale in long-form video: **VideoAgent** processes 20× fewer frames while improving accuracy, and **VLog**'s 124M model matches 7B baselines through generative retrieval, demonstrating that the inherent sparsity of task-relevant information rewards efficiency-aware architectures.

### 5.4 Multimedia Content Generation and Editing

Text-based tool orchestration approaches demonstrate compositional flexibility but cannot leverage visual feedback for quality assessment. VisProg (Tanmay Gupta, 2022) generates Python-like programs that compose off-the-shelf vision modules, achieving strong zero-shot performance on compositional reasoning tasks. However, the architecture lacks mechanisms to verify whether the generated outputs align with user intent, thereby limiting its applicability to iterative creative workflows. Native multimodal agents address this through direct visual perception for both tool selection and output verification. GenArtist (Wang et al., 2024f) employs GPT-4V to perceive generated images and perform self-correction through a planning tree mechanism. The visual feedback loop enables the detection of attribute-binding failures and spatial relationship errors that text-based systems cannot identify. This architectural choice yields an over 20% improvement in spatial relationship accuracy compared to DALL-E 3 on T2I-CompBench, with an additional 7-10% improvement attributable to the hierarchical planning tree over simpler chain-based approaches. The results demonstrate that native visual perception enables qualitatively distinct error-correction capabilities, in which iterative refinement guided by visual verification yields outputs unattainable through single-pass generation. CREA (Venkatesh et al., 2025) extends this principle from single-agent self-correction to multi-agent collaborative refinement. By decomposing the creative process into specialized roles, including a Creative Director, an Art Critic, and a Refinement Strategist, CREA enables iterative feedback loops that address multiple quality dimensions simultaneously, outperforming state-of-the-art methods in diversity, semantic alignment, and creative transformation.

The subjective nature of multimedia generation and editing makes it inherently difficult to establish definitive success and failure criteria. Consequently, the evaluation of multimodal agents in this domain is frequently hindered by a lack of comprehensive, real-world benchmarks, prompting many researchers to manually curate datasets tailored to their specific tasks. A primary challenge is that most existing benchmarks lack realistic, multi-turn editing samples and ground truths. While MagicBrush stands out as one of the few large-scale benchmarks in this category, it is widely noted that alternative datasets are often too noisy or overly domain-specific. Furthermore, standard benchmarks are generally insufficient for agentic systems tasked with handling complex, composite instructions. To bridge this gap, frameworks such as CoSTA* and FaSTA* have developed custom benchmarks specifically to evaluate cost-sensitive multimodal generation and editing. Similarly, while systems like VISPROG and CLOVA rely on standard datasets like GQA for compositional and grounded visual reasoning, their authors have had to manually collect instructions and images to adequately evaluate their image editing capabilities.

> **Note:** Native visual perception enables qualitatively distinct capabilities in content generation: **GenArtist**'s visual feedback loop achieves 20%+ improvement in spatial relationship accuracy through iterative self-correction, a capability fundamentally unavailable to text-mediated pipelines that lack output verification mechanisms.

Across application domains, the architectural trajectory follows a consistent pattern: tighter integration between perception and reasoning reduces information bottlenecks that constrain earlier approaches. Early-fusion architectures in robotics yield 20-30 percentage point improvements through preserved visual semantics and unified action tokenization. Selective attention mechanisms in video understanding achieve substantial efficiency gains without sacrificing accuracy by aligning computational allocation with task-relevant information density. Visual feedback loops in content generation enable iterative refinement cycles impossible in text-mediated pipelines. However, the persistent 60-70 percentage point gap between model and human performance on GUI benchmarks indicates that current architectures, regardless of fusion strategy, have not yet achieved the grounding precision, planning depth, and failure recovery mechanisms required for human-level performance on complex interactive tasks.

## 6 Efficiency

The efficiency of multimodal agentic frameworks is defined by their ability to process high-bandwidth sensory data and execute complex tasks within the temporal constraints of their operating environments. Unlike text-only agents, multimodal systems must manage the computational overhead of high-resolution visual

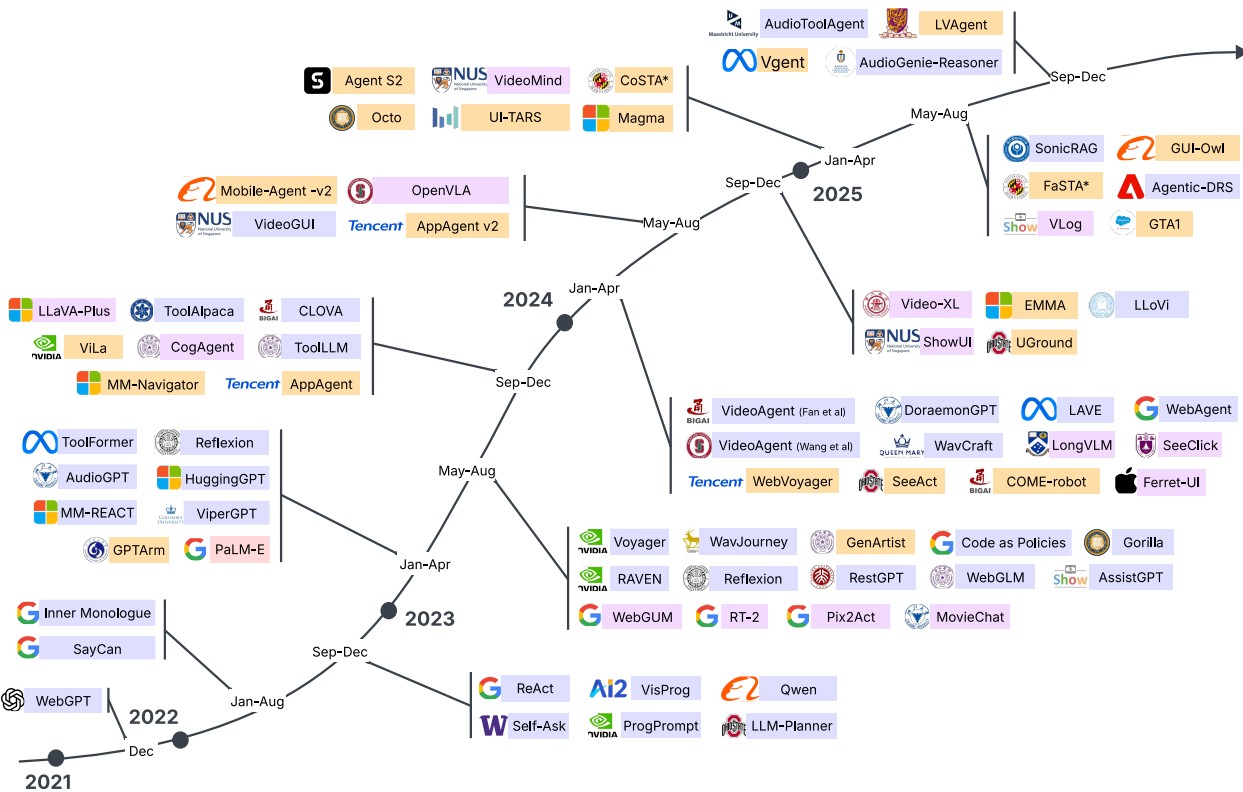

Figure 4: **Evolution of Agentic Frameworks:** This diagram shows the evolution of agentic frameworks, starting from late 2021 with WebGPT, which was one of the primitive works. The different agents are distinguished by the perception technique they use: delegated , late-fusion , and early-fusion . This aligns with the approach used throughout the paper for comparing frameworks across different application areas.

encoding (Dosovitskiy, 2020; Radford et al., 2021a), temporal video modeling (Shen et al., 2024; Shu et al., 2024), or continuous audio processing (Elizalde et al., 2022; Radford et al., 2022) alongside the orchestrator's reasoning cycles. The efficiency of these agentic frameworks is primarily analyzed through (i) inference latency, (ii) throughput, and (iii) memory overhead. These metrics reveal the trade-offs between the modularity of API-based tool-use agents and the low-latency response times of optimized, fine-tuned models.

## 6.1 Robotics and Physical Embodiment

**LLM-based Tool-Use Agents** The efficiency of these systems is fundamentally constrained by their "stop-and-think" operational paradigm, where the robot must pause to query a remote Large Language Model (LLM) for high-level planning before executing low-level skills. Code as Policies (Liang et al., 2023) generates Python code to link perception APIs with control primitives, making the system's response time dependent on both the code-generation latency of models such as Codex and the execution time of the generated script. Similarly, SayCan (Ahn et al., 2022) and Inner Monologue (Huang et al., 2022) rely on iterative queries to LLM APIs (e.g., GPT-3) to score affordances or process textual feedback, which limits control frequency compared to end-to-end policies and introduces network latency bottlenecks. Consequently, these agents typically function as high-level semantic planners with lower throughput rather than high-frequency motor controllers.

**Fine-tuned Late-fusion Multimodal Agents** Inference latency is a critical bottleneck for these Vision-Language-Action (VLA) models due to the sheer size of their integrated backbones. RT-2 (Ichter et al., 2023), for instance, employs models up to 55B parameters, necessitating multi-TPU cloud inference to achieve a control frequency of only 1–3 Hz, while a smaller 5B variant can reach approximately 5 Hz. To address

these latency hurdles, OpenVLA (Kim et al., 2024) employs a smaller 7B backbone and 4-bit quantization to achieve an inference throughput of approximately 6 Hz on a single consumer-grade GPU. Octo (Octo Model Team et al., 2024) further prioritizes efficiency by offering smaller checkpoints (e.g., 27M and 93M parameters) that support faster inference and easier deployment compared to their multi-billion parameter counterparts.

**Natively Multimodal Agents** The efficiency of natively multimodal agents is often hampered by the high inference latency inherent in large foundation models, which limits their applicability in real-time closed-loop control. GPTArm (Zhang et al., 2025a) exhibits significant latency due to its reliance on the GPT-4V API, with task execution times averaging 33-110 seconds, which forces a distinct "stop-and-think" behavior. Similarly, COME-robot (Zhi et al., 2025a) employs a closed-loop feedback mechanism that verifies status and restores plans via GPT-4V; while this enhances robustness, the iterative check-and-act cycles inevitably lower system throughput. EMMA (Hwang et al., 2024), built on Gemini, addresses onboard deployment challenges by optimizing the end-to-end model to remove explicit reasoning chains, achieving an inference speed of approximately 3 Hz. $\pi_0$ (Black et al., 2024) achieves a qualitative leap in control frequency through its action-chunking flow matching design, reaching up to 50 Hz, which represents an order-of-magnitude improvement over the 1-3 Hz of RT-2 and the 6 Hz of OpenVLA, enabling closed-loop dexterous manipulation that prior VLA architectures could not support.

> **Note:** *Fine-tuned late-fusion* multimodal agents like **Octo** and **OpenVLA** offer the most practical solution for efficiency by providing optimized architectural backbones and quantization to facilitate real-time deployment on consumer hardware, thereby mitigating the inference latency bottleneck.

## 6.2 GUI and Web Navigation

**LLM-based Tool-Use Agents** The efficiency of text-based agents has evolved significantly, largely revolving around the trade-off between parsing overhead and inference speed. Early iterations, such as WebGPT (Reiichiro Nakano et al., 2021), exhibited severe latency, taking approximately 52 seconds per query due to the sequential nature of browsing and the model size of 175 B. However, subsequent systems, such as WebGLM (Liu et al., 2023a), optimized this process considerably, achieving an average response time of approximately 5.36 seconds, nearly 10 times faster than WebGPT. To improve throughput, AutoDroid (Wen et al., 2024) prunes the UI representation by merging functionally equivalent elements, reducing the token count by nearly half and thereby accelerating inference.

**Fine-tuned Late-fusion Multimodal Agents** These agents focus on efficient processing of high-resolution visual inputs to overcome the latency bottlenecks associated with traditional visual encoding. CogAgent (Hong et al., 2024) introduces a high-resolution cross-module that runs in parallel with a low-resolution branch, a design that allows the model to process 1120x1120 images with FLOPs (floating-point operations) increasing linearly rather than quadratically, making high-resolution processing computationally feasible and faster. SeeClick (Cheng et al., 2024) further reduces response times by directly predicting action locations from screenshots, thereby bypassing the latency associated with parsing structured text or HTML, and is particularly efficient on platforms where metadata is unavailable or verbose. In order to manage the high computational cost and memory usage associated with processing high-resolution GUI screenshots over long trajectories, GUI-Owl (Ye et al., 2025) employs a strategy of retaining only the most recent 1 to 3 images in the context window, which saves GPU memory and improves inference speed.

**Natively Multimodal Agents** The efficiency of natively multimodal agents is often constrained by the latency of processing heavy visual data through API calls. Mobile-Agent (Wang et al., 2024c) faces significant latency because it must capture a screenshot, upload it to the GPT-4V API, and wait for a response at every step of the operation. To address throughput issues caused by long-context histories in sequential tasks, Mobile-Agent-v2 (Wang et al., 2024b) adopts a multi-agent architecture in which a *planning agent* condenses the history into a text-based task-progress summary. This reduces the context length passed to the *decision agent*, thereby speeding up decision-making compared to single-agent architectures that process full interleaved image-text histories. In AppAgent (Zhang et al., 2023a), the framework improves efficiency by creating a reference document during the exploration phase and using it as long-term memory

for the agent to consult during new-task execution. AppAgent v2 (Li et al., 2024) further improves memory retrieval efficiency by employing Retrieval-Augmented Generation (RAG) (Lewis et al., 2020). UI-TARS-2 (Wang et al., 2025a) addresses the data scalability bottleneck through an automated data flywheel that continuously collects and reflectively refines interaction traces, enabling self-improvement without expensive manual annotation. TON (Wang et al., 2026) targets inference cost directly by training the VLM to decide when reasoning is even necessary, using a "thought dropout" stage followed by reinforcement learning so that easy steps skip explicit reasoning traces entirely—cutting generated token length substantially without degrading task performance. Agent S2 (Agashe et al., 2025) reduces redundant computation by proactively refining plans before execution through its hierarchical planning mechanism, rather than relying on costly reactive error correction after failed actions.

> **Note:** While natively multimodal agents are training-free and easy to set up, they suffer from high latency and prohibitive token costs due to API calls. Fine-tuned multimodal agents such as **GUI-Owl** and **CogAgent** appear to deliver superior inference speed, lower operational costs, and higher precision for practical, scalable deployment.

### 6.3 Multimedia Content Generation and Editing

**LLM-based Tool Use Agents** These multimodal tool-use agents achieve efficiency by moving intelligence into planning and relying on selective tool execution instead of heavy end-to-end training. VISPROG (Tanmay Gupta, 2022) illustrates this by using an LLM to convert a visual query into a symbolic program that calls pretrained vision models and simple operations, avoiding training and running only the components needed per query. AudioGPT (Huang et al., 2023a) follows the same pattern in audio, where an LLM coordinates existing speech, music, and sound models through modality interfaces, supporting many tasks without a unified multimodal model. WavJourney (Liu et al., 2023b) and WavCraft (Liang et al., 2024) extend this idea by having the LLM generate structured scripts or executable code that trigger specialized audio generators and editors, encoding temporal and semantic structure while avoiding expensive joint optimization; despite this modular design, WavCraft (Liang et al., 2024) matches or surpasses end-to-end baselines. LAVE (Wang et al., 2024a) introduces the planner–executor paradigm to video editing, leveraging language to retrieve clips and plan edits, thereby reducing user effort and workflow complexity.

**Natively Multimodal Agents** These systems achieve efficiency by reasoning directly over images and videos while coordinating pretrained models through agent-style planning, instead of retraining new generators. GenArtist (Wang et al., 2024f) uses a multimodal LLM to decompose complex instructions, select suitable generation or editing tools, and verify outputs, enabling both image generation and editing without end-to-end training and achieving strong compositional gains. CoSTA* (Gupta et al., 2025b) shows that efficiency can also come from prompting itself, providing a systematic taxonomy that identifies prompt strategies that improve performance with minimal inference overhead and no training. FaSTA* (Gupta et al., 2025a) applies fast–slow planning to multi-turn image editing, relying on quick LLM plans for common cases and invoking expensive search only when needed, while reusing past toolpaths to reduce repeated cost. UniEdit (Bai et al., 2025a) and AnyV2V (Ku et al., 2024) extend this tuning-free philosophy to video editing, using pretrained diffusion and video models to apply and propagate edits without fine-tuning. Overall, these works show that efficiency comes from selective computation, plan-driven control, and reuse of pretrained models rather than costly retraining. CREA (Venkatesh et al., 2025) incurs additional overhead from its iterative multi-agent debate loop, as each refinement cycle involves sequential LLM calls for planning, critique, and scoring alongside diffusion model inference, representing a trade-off between creative quality and latency analogous to the cost structure observed in multi-turn planning frameworks such as FaSTA* (Gupta et al., 2025a).

> **Note:** While delegated perception frameworks like **VISPROG** and **AudioGPT** are clear winners when it comes to efficiency on simple generation and editing tasks, they perform poorly on multi-turn complex editing. Therefore, natively multimodal agents such as **FaSTA\***, which employ novel strategies for cost and latency such as hierarchical planning, are the way forward.

## 6.4 Long-Form Video Understanding and Retrieval

**LLM-based Tool Use Agents** Early approaches to long-form video understanding relied on dense frame-level processing, which quickly became computationally prohibitive as video length increased. Recent agentic methods instead treat video understanding as a guided exploration problem, where only the most relevant visual evidence is processed. VideoAgent (Zhi et al., 2025b) illustrates this shift by using an LLM to iteratively retrieve task-specific frames, reducing visual input to 8 frames per query on benchmarks such as EgoSchema (Zhuang et al., 2023) and NExT-QA (Xiao et al., 2021) while maintaining strong zero-shot accuracy (54.1% and 71.3%, respectively). This stands in sharp contrast to dense baselines that process hundreds of frames per query. LLoVi (Zhang et al., 2024) takes a different but complementary route, sidestepping long-range visual modeling entirely by first converting videos into short textual summaries and then reasoning purely in the language space; by sampling 8× fewer clips, it achieves comparable performance with only a 2% accuracy drop. Building on this abstraction-first philosophy, DoraemonGPT (Huang et al., 2025) further compresses visual input into a symbolic spatio-temporal memory and uses an MCTS-based planner to guide reasoning, selectively avoiding unnecessary computation. Together, these works trace a clear evolution from brute-force video encoding toward agent-driven retrieval, abstraction, and planning as the foundation for efficient long-horizon video reasoning.

**Fine-tuned Late-fusion Multimodal Agents** Efficiency is achieved by carefully modifying specific components rather than fully fusing vision and language at the token level. VideoMind (Liu et al., 2025) improves efficiency by using a Chain-of-LoRA (Wenhan Xia, 2024) design on top of a Qwen2-VL (Bai et al., 2024) backbone, where planning, grounding, and verification are implemented as lightweight LoRA (Hu et al., 2021) adapters. This avoids running multiple full models for different reasoning roles, allows the model to switch roles during inference, and reduces memory and inference cost while maintaining strong performance on long-video benchmarks. VLog (Lin & Shou, 2025) addresses efficiency from the generation side by replacing slow token-by-token text generation with generative retrieval over a compact, hierarchically organized event vocabulary. By retrieving minimal narration units rather than generating full-text sequences, VLog significantly reduces decoding overhead and achieves substantial speedups on long videos without compromising accuracy. Together, these works show that late-fusion agents can scale to long-video reasoning by compressing either the reasoning modules or the generation space, rather than processing dense token sequences.

**Natively Multimodal Agents** Multimodal Efficiency in native multimodal agents is increasingly defined by their ability to reason over long, multimodal video streams without prohibitive compute cost. The Video-MME (Fu et al., 2024) benchmark reveals that even strong commercial MLLMs achieve only about 75% accuracy and exhibit a clear performance drop as video length increases, exposing inefficiencies in long-sequence visual reasoning. However, adding subtitles and audio consistently improves results, showing that richer modality fusion enhances efficiency. Gemini 1.5 (Gemini Team, 2024) addresses these challenges by scaling multimodal context to millions of tokens, achieving near-perfect long-context retrieval (>99% up to at least 10M tokens) and strong gains on long-video QA while reducing training and serving compute, especially in the Flash variant, leading to 26–75% real-world time savings. Complementarily, GPT-4o (OpenAI, 2024) improves efficiency through a unified end-to-end omni-modal architecture, delivering human-like response latency ($\simeq$232 ms for audio) and roughly 50% lower serving cost than GPT-4 Turbo (OpenAI, 2023) without sacrificing performance. Beyond scaling commercial models, a parallel line of work targets efficiency at the architecture level for streaming video. VideoLLM-online (Chen et al., 2024a) introduces a Learning-In-Video-Stream framework that enables temporally aligned, real-time dialogue over continuous video, with an inference pipeline that parallelizes encoding and generation to reach over 10 FPS on a single GPU. VideoLLM-MoD (Wu et al., 2024b) further cuts the dominant cost of dense vision tokens by applying a mixture-of-depths strategy that learns to skip computation for a large fraction of vision tokens at certain transformer layers, yielding roughly 42% time and 30% memory savings while preserving or improving accuracy. VideoRAG (Ren et al., 2025) enables processing of hundreds of hours of video on a single consumer GPU by distilling content into structured knowledge representations and applying graph-based multi-modal retrieval rather than dense encoding. VideoDeepResearch (Yuan et al., 2025) demonstrates superior cost-effectiveness by progressively reasoning over expanding frame subsets, consuming fewer total visual tokens than both single-pass MLLMs and static retrieval baselines.

**Note:** *Delegated* and *late-fusion* are the most efficient strategies, as they sidestep the quadratic complexity of full-sequence processing by leveraging the inherent sparsity of task-relevant information. We observe this in frameworks such as **VideoAgent**, which processes 20× fewer frames, and **VLog**, which achieves 10-20× speedup due to generative retrieval.

## 7 Scalability

Scalability examines how agentic frameworks adapt as task complexity, data volume, and environmental diversity increase. For multimodal agents, this involves a critical trade-off between training compute (upfront infrastructure costs) and operational costs (recurring per-token API fees). We determine scalability by evaluating (i) data efficiency, which involves the model's ability to generalize to novel tasks through few-shot prompting (Brown et al., 2020; Wei et al., 2022c) or parameter-efficient fine-tuning (Hu et al., 2021; Wenhan Xia, 2024), (ii) modularity, which takes into account how easily new modalities or tools can be integrated without retraining the entire system (Qin et al., 2024; Alayrac et al., 2022) and (iii) deployment costs, where we study the sustainability of proprietary API dependencies (OpenAI, 2023; Gemini Team, 2024) versus locally-hosted and domain-specific models (Hong et al., 2024; Lai et al., 2024).

### 7.1 Robotics and Physical Embodiment

**LLM-based Tool-Use Agents** The scaling of these agents is primarily constrained by operational costs and the availability of robust low-level primitives, rather than the training compute of the agent itself. Because these frameworks utilize proprietary LLMs such as GPT-3 (Brown et al., 2020), Codex (Chen, 2021), or InstructGPT (Ouyang et al., 2022) via APIs, they incur recurring token costs for every plan generated or line of code synthesized. However, they avoid the massive data collection and compute required for end-to-end training. Code as Policies (Liang et al., 2023), for instance, does not require training on a fixed set of predefined skills or policies as it can directly generate policy code. Similarly, Inner Monologue (Huang et al., 2022) operates via few-shot prompting of the reasoning model, allowing it to scale to new tasks as long as the underlying perception tools and robotic skills are available.

**Fine-tuned Late-fusion Multimodal Agents** Training these models requires massive computational resources; OpenVLA (Kim et al., 2024) required 21,500 A100 hours for pre-training, and PaLM-E (Danny Driess et al., 2023) scales up to 562 billion parameters, demanding extensive distributed infrastructure. Magma (Yang et al., 2025) also demonstrates the data-hungry nature of this category, utilizing a curated dataset of approximately 39 million diverse samples to achieve broad spatial-temporal intelligence. Despite high initial costs, these models offer scalable fine-tuning mechanisms: Octo (Octo Model Team et al., 2024) enables efficient fine-tuning to new observation and action spaces within a few hours, and Open-VLA can be adapted to new tasks using Low-Rank Adaptation (LoRA) on consumer hardware, significantly reducing the memory footprint required for adaptation.

**Natively Multimodal Agents** Scaling dynamics for this category diverge based on whether the model is accessed via API or trained end-to-end. Systems such as GPTArm (Zhang et al., 2025a) and COME-robot (Zhi et al., 2025a) rely on proprietary APIs, which avoid training compute but incur recurring costs for each visual query and planning step. In contrast, EMMA (Hwang et al., 2024) is fine-tuned end-to-end, which requires substantial training compute to align the foundation model with driving tasks but allows the model to internalize world knowledge for autonomous operation without per-token fees during deployment. However, deploying such large models (e.g., Gemini or PaLI) on board vehicles or robots remains a scaling challenge due to the high computational requirements for inference.

**Note:** In the domain of Robotics and Physical Embodiment, *fine-tuned late-fusion* multimodal agents like **Octo** and **OpenVLA** appear most viable for future scaling because they effectively transfer semantic knowledge into generalized robotic control, while recent advances in parameter-efficient fine-tuning and quantization are successfully mitigating their computational bottlenecks.

## 7.2 GUI and Web Navigation

**LLM-based Tool Use Agents** Scaling these agents involves balancing inference compute against API costs. WebAgent (Gur et al., 2023) addresses inference-compute limitations by employing a domain-expert model, such as HTML-T5 (Gür et al., 2023), for planning. This approach is more computationally efficient than relying on a large general-purpose LLM for each substep of a task. Similarly, AutoWebGLM (Lai et al., 2024) is designed for practical scalability, using ChatGLM3-6B (GLM et al., 2024), a 6-billion-parameter model, thereby reducing inference costs relative to larger GPT-4-based baselines while maintaining high performance. However, systems such as AssistGUI (Gao et al., 2023a) and AutoDroid (Wen et al., 2024) (when using GPT-4) face scalability challenges related to cost, as they must submit dense textual descriptions of the screen state to paid APIs at every step, motivating optimizations to reduce prompt length to reduce these recurring expenses.

**Fine-tuned Late-fusion Multimodal Agents** The scaling profile for this category is characterized by high initial training costs but favorable inference economics. Training models like CogAgent (Hong et al., 2024) and GUI-Owl (Ye et al., 2025) require massive compute resources; for instance, CogAgent was pre-trained on datasets like LAION-2B and COYO-700M for 60,000 iterations, and GUI-Owl utilizes a large-scale cloud infrastructure to generate sufficient training data. However, once trained, inference is highly scalable; CogAgent-18B and GUI-Owl-7B achieve better performance than much larger generalist models on GUI tasks, offering a superior performance-to-parameter ratio that makes them more cost-effective for widespread deployment.

**Natively Multimodal Agents** Scaling these systems is primarily limited by high operational API costs rather than training compute. AppAgent (Zhang et al., 2023a) and Mobile-Agent (Wang et al., 2024c) incur costs for every screenshot processed, which can become prohibitive for long-horizon tasks. To improve scalability, AppAgent v2 (Li et al., 2024) employs Retrieval-Augmented Generation (RAG) to dynamically retrieve and update a knowledge base, thereby avoiding the need to feed the entire interaction history into the context window and optimizing token usage. While they avoid the massive compute costs of pre-training, frameworks such as AppAgent incur an initial *exploration cost* to build reference documents before they can be deployed efficiently.

**Note:** The scalability of these three categories of frameworks clearly indicates that it would be more prudent to accept some high upfront infrastructure cost, as in **GUI-Owl** and **CogAgent**, than rely on expensive, token-heavy proprietary APIs that incur costs with every call.

## 7.3 Multimedia Content Generation and Editing

**LLM-based Tool Use Agents.** Scalability in these agents comes from avoiding large end-to-end training and instead relying on modular, reusable components. VISPROG (Tanmay Gupta, 2022) scales visual reasoning by converting each query into a small program that calls only the required vision tools, so harder or longer reasoning chains do not increase training cost or model size. AudioGPT (Huang et al., 2023a) follows the same idea for audio: tasks are handled by selecting and combining existing speech, music, or sound models, and new capabilities are added by plugging in models rather than retraining the system. WavJourney (Liu et al., 2023b) and WavCraft (Liang et al., 2024) scale to complex audio creation and editing by using LLMs to break high-level instructions into executable steps over specialized audio modules, enabling richer compositions without training new networks. LAVE (Wang et al., 2024a) applies this principle to video editing, where an LLM plans editing actions from language descriptions of videos, allowing new editing goals without task-specific retraining. AssistEditor (Gao et al., 2024a) extends this paradigm to professional editing software, using a collaborative multi-agent framework where specialized GUI agents translate high-level user intent into actions that control video models and tools such as Premiere Pro. MovieBench (Wu et al., 2025) complements these efforts with a hierarchical, movie-level dataset that enables evaluation of long video generation with consistent characters and coherent multi-scene storylines. Overall, these works show that scalability arises from modular execution and the reuse of existing models, thereby keeping computational and training costs stable as task diversity and complexity increase.

**Natively Multimodal Agents.** These agents scale by planning over images and videos while reusing existing models, rather than retraining or enlarging architectures. GenArtist (Wang et al., 2024f) handles diverse image generation and editing tasks by decomposing complex prompts into tool-level plans and coordinating existing generators with self-correction. FaSTA* (Gupta et al., 2025a) improves multi-turn image editing efficiency through fast–slow planning, reusing frequent tool subroutines, and invoking costly search only when needed. UniEdit (Bai et al., 2025a) scales video editing with a tuning-free framework that reuses a pretrained text-to-video backbone with modular motion and appearance branches. AnyV2V (Ku et al., 2024) further enables scalable video editing through a plug-and-play design that integrates image editors with image-to-video models, supporting a wide range of tasks without retraining.

**Note:** As most agentic frameworks in this domain are tool-use in nature, scalability is defined by how robustly and economically they can handle complex multi-turn editing. While LLM-based agents have lower token costs, frameworks such as **FaSTA*** and **AnyV2V**, which employ efficient editing approaches, are the way forward.

### 7.4 Long-Form Video Understanding and Retrieval

**LLM-based Tool Use Agents** These agents scale long-video understanding by avoiding dense frame processing and relying on selective, modular reasoning. VideoAgent (Zhi et al., 2025b) retrieves only task-relevant frames ($\approx$8.4 per query on average), allowing efficient reasoning over hour-long videos without retraining. EgoSchema (Zhuang et al., 2023) and NExT-QA (Xiao et al., 2021) support scalability by evaluating long-range temporal and causal reasoning at scale, exposing the limits of full-sequence processing. LLoVi (Zhang et al., 2024) converts long videos into compact captions and lets an LLM perform long-range reasoning, avoiding heavy temporal modeling. DoraemonGPT (Huang et al., 2025) further scales to dynamic scenes by compressing videos into symbolic memories and using modular tools with MCTS-based planning. Overall, these works show that scalability in video agents arises from retrieval, abstraction, and planning rather than from processing entire videos end-to-end.

**Fine-tuned Late-fusion Multimodal Agents** These multimodal agents scale by reusing a single backbone model with lightweight adaptations instead of multiple heavy networks. VideoMind (Liu et al., 2025) employs a role-based workflow implemented via Chain-of-LoRA (Wenhan Xia, 2024), activating Planner, Grounder, Verifier, and Answerer roles via small LoRA adapters within a single model, enabling efficient role switching across long videos. Chain-of-LoRA (Wenhan Xia, 2024) increases capacity by stacking and merging LoRA (Hu et al., 2021) modules without increasing inference cost, thereby keeping training efficient as tasks grow. Qwen2-VL (Bai et al., 2024) supports scaling through dynamic resolution handling and unified multimodal position embeddings, showing steady gains as model and data size increase. VLog (Lin & Shou, 2025)further improves scalability by compressing long videos into reusable textual narrations, reducing per-frame processing. Overall, these works show that late-fusion agents scale through parameter-efficient tuning, compact representations, and shared backbones.

**Natively Multimodal Agents** These agents scale by directly processing very large multimodal contexts within a single unified model. Gemini 1.5 Pro (Gemini Team, 2024) demonstrates long-context scalability by reasoning over millions of tokens across text, video, and audio, enabling the processing of entire books or hours of video without chunking or external retrieval. GPT-4o (OpenAI, 2024) scales across modalities within a single end-to-end architecture that integrates text, vision, audio, and video, achieving fast responses and lower cost without separate pipelines or models. From the evaluation side, Video-MME (Fu et al., 2024) enables scalable assessment of video understanding across short to hour-long videos and multiple modalities, showing how performance changes as sequence length grows and highlighting the need for efficient long-context reasoning. Together, these works show that native agents scale through unified architectures and long-context processing rather than modular decomposition or task-specific training.

**Note:** In long-form video understanding, scalability means efficiently summarizing and reasoning over key temporal cues rather than processing every frame. This is evident in tool use and fine-tuned frameworks such as **VideoAgent**, **VideoMind**, and **VLog**.

# 8 Latency

To address the significant latency bottlenecks affecting multimodal agentic systems, several novel frameworks have come up with targeted strategies ranging from prompt optimization to architectural innovations. Approaches like prompt pruning and token reduction are highly effective in reducing the number of model calls and context. Latency requirements are also dependent on the application area where a system is being deployed. While domains such as Robotics and Web/GUI navigation strictly require a low-latency approach, other end-to-end domains like multimodal understanding and editing are more forgiving towards higher latency frameworks.

## 8.1 Robotics and Physical Embodiment

For systems like GPTArm (Zhang et al., 2025a) that rely heavily on VLM API calls, strategies such as adaptive feedback frequencies have been employed. This controls the number of times the intermediate results of the framework have to be evaluated by a VLM. It was observed that inference latency dropped significantly, especially for complex tasks, by ∼50%. While there was a minimal performance drop in simple tasks, the reduced evaluation steps resulted in a ∼9% drop in SR for complex tasks. Frameworks that use end-to-end models like EMMA (Hwang et al., 2024) have benefited from simple solutions like bypassing the generation of CoT reasoning steps, which has improved inference latency by 67%. Other systems like RT-2 (Ichter et al., 2023) have chosen to offload the heavy inference of their 55B models to distributed cloud services using multi-TPU hardware so that they can achieve a viable control frequency of 1 to 3 Hz, even with their largest models. Alternatively, approaches like Octo (Octo Model Team et al., 2024) utilize much smaller, highly efficient backbones (27 - 93M) to process inputs locally and rapidly without the overhead of massive foundation models. OpenVLA (Kim et al., 2024), while keeping a 7B-parameter model, has increased its throughput using 4-bit quantization. This allows it to run at ∼3Hz on an NVIDIA A5000 GPU.

Despite these optimizations, deploying these agents in high-frequency, safety-critical production environments remains extremely challenging due to a trade-off between speed and reliability. While several frameworks like EMMA and OpenVLA have managed to achieve single-digit control frequencies with powerful VLMs, real-world environments often require significantly higher rates for smooth motor control. This almost eliminates the possibility of using external API-based model calls, as they introduce network dependencies and lead to unfeasible interruptions. On the other hand, systems that reduce their visual verification frequency to improve speed, like GPTArm, become vulnerable to mid-task anomalies and execution failures. Additionally, frameworks such as EMMA can face deployment hurdles due to a large context requirement that comes from fusing LiDAR/radar data with continuous camera streams. Such large context windows can significantly increase inference latency.

## 8.2 GUI and Web Navigation

AutoDroid Wen et al. (2024) reduces inference latency by merging functionally equivalent UI elements to cut down prompt token count by almost 50%. It also minimizes the number of LLM calls by leveraging shortcuts from memory, built during an offline exploration phase. These implementations bring down inference latency by 21.3%. Meanwhile, WebAgent (Gur et al., 2023) deploys a lightweight HTML-T5 (Gür et al., 2023) model to summarize long web pages before passing them to an LLM. CogAgent (Hong et al., 2024) takes a different approach by using a locally deployed VLM that eliminates the network-induced latency and cost bottlenecks associated with querying large API-based models. However, to reduce inference latency, they implement a cross-attention module between dual branches of low and high-resolution features, making the processing complexity linear with respect to visual features. This provides a massive reduction in computational cost (∼25x), which can significantly improve inference latency. Frameworks such as WebGLM (Liu et al., 2023a) speed up inference by reducing the latency of webpage retrieval from ∼2 mins to 5 secs using parallel asynchronous crawling.

Despite these optimizations, a primary hurdle for seamless deployment of these systems is the heavy dependence on proprietary APIs or large local models, where network instability and intensive compute requirements create "stop-and-think" delays that throttle real-time action frequency. Furthermore, frameworks like

WebVoyager (He et al., 2024a) frequently fail by getting trapped in navigation loops or hallucinating when faced with extremely long contexts, which is quite common in web navigation. This error accumulation is made worse by the grounding bottleneck that is prevalent in modern VLMs. It has been observed in frameworks such as SeeAct (Zheng et al., 2024) and MM-Navigator (Yan et al., 2023), where agents struggle to translate a high-level text intention into exact, localized UI coordinates. Finally, the highly dynamic nature of the live internet environment, due to its frequent pop-up interruptions, floating ads, and unpredictable loading states, forces agents to constantly employ self-reflection and state verification to avoid errors. This has been extensively studied in AutoWebGLM (Lai et al., 2024), where the authors have admitted that such concerns make it difficult to deploy web agents seamlessly.

# 9 Reliability

Multimodal agentic systems need to perform consistently in real-world conditions to be feasible for production deployment. These conditions include out-of-distribution data, unexpected intermediate outputs, and handling of cascading errors. Unlike latency, which produces a measurable delay, reliability failures are often difficult to identify. They could involve a hallucinated bounding box, an incorrectly captioned spatial relationship, or an outdated state representation. Several frameworks across application domains have introduced targeted mechanisms to contain these failure modes, though significant gaps remain before reliable production deployment is achievable.

## 9.1 Robotics and Physical Embodiment

To address the cascading error problem inherent in long-horizon physical task execution, several frameworks like Inner Monologue (Huang et al., 2022) introduce closed-loop verification between planning steps. After each physical action, the agent re-queries its perception tools to confirm the resulting state before generating the next subtask plan. If the observed state diverges from the expected state, the agent replans from the verified configuration rather than continuing on an outdated premise. COME-robot (Zhi et al., 2025a) extends this with a dual-loop architecture, where an inner verification loop continuously monitors subtask outcomes and triggers replanning in the outer task-decomposition loop when failures are detected. SayCan (Ahn et al., 2022) improves planning-level reliability through affordance scoring by evaluating both the semantic validity and physical feasibility of proposed actions before execution. This reduces the rate of physically incoherent plans from reaching the robot's actuators. At the model level, end-to-end VLA frameworks such as RT-2 (Ichter et al., 2023) and OpenVLA (Kim et al., 2024) improve reliability by eliminating the loss that is usually associated with modular approaches. A unified perception-to-action framework enables the spatial and semantic information to be preserved.

Despite these advances, deploying robotic agents reliably in production environments remains fundamentally limited by the tradeoff between verification frequency and control speed. Frameworks that perform dense visual verification, such as COME-robot and GPTArm (Zhang et al., 2025a), introduce remote API calls at each subtask level. This improves recovery but reduces control frequency to a range that is impractical for real-world manipulation tasks. Conversely, reducing verification frequency to improve inference speed, as observed in GPTArm, results in approximately a 9% drop in success rate on complex tasks. This shows that reliability and throughput are structurally opposed in current frameworks. VLA models, despite their end-to-end integration, exhibit their own reliability failure mode: they are trained on fixed physical configurations and degrade unpredictably when deployed on unseen platforms and data. Octo (Octo Model Team et al., 2024) partially addresses this by training a single, unified robot policy, trained on data collected from diverse configurations. However, out-of-distribution generalization remains an open problem.

## 9.2 GUI Grounding and Web Navigation

Reliability in the GUI/web navigation domain is easier to track because a single wrong click or incorrect element identification can immediately produce a detectable incorrect screen state. SeeAct (Zheng et al., 2024) addresses this grounding bottleneck with a two-pass verification strategy. Once it has identified the target element, the agent overlays SoM tags on the screenshot and re-queries GPT-4V to confirm the correct

element index before finalizing the action. WebVoyager (He et al., 2024a) and Mobile-Agent (Wang et al., 2024c) have adopted a post-action verification strategy, where they compare the resulting screen state after an action against the expected state. If the observed state does not match the ground truth, the agent triggers a correction. In addition to this, Mobile-Agent-v2 (Wang et al., 2024b) addresses the context degradation problem that affects all long-horizon tasks by introducing a dedicated planning agent that continuously compresses the interaction history into a text-based progress summary. Then there are approaches like AppAgent-v2 (Li et al., 2024), which use an offline exploration phase to build a reference document of app behaviors, which is then retrieved via RAG at inference time. This reduces the agent's reliance on zero-shot generalization to novel GUI states.

Despite these mechanisms, reliable production deployment of GUI agents remains significantly constrained due to the randomness of live digital environments. The authors of WebVoyager (He et al., 2024a) have shown that the agent often fails when it is provided with unexpectedly long context, as it gets trapped in navigation loops. The requirement of continuous self-reflection and state verification becomes necessary due to the internet's frequent interruptions, floating advertisements, and unpredictable loading states. This creates a barrier to production deployment, as seen in AutoWebGLM (Lai et al., 2024) While frameworks such as SeeAct (Zheng et al., 2024) and MM-Navigator (Yan et al., 2023) have identified grounding bottlenecks, they have also shown that current SOTA VLMs often struggle to translate high-level semantic intent into precise pixel-level coordinates, even with SoM overlays. Code2World (Zheng et al., 2026) gives agents predictive foresight by simulating the next interface state through renderable code generation, allowing candidate actions to be evaluated before execution rather than verified only afterward. This is reflected in the persistent 60-70% gap between multimodal agent performance and human performance on WebArena benchmarks, which evaluate precise, multi-step, error-recovering interaction that production deployments require.

## 10   Limitations

As detailed in our preceding analyses of performance, efficiency, and reliability, eliminating the information loss inherent in text conversion does not automatically resolve the deeper cognitive and computational challenges of multimodal agency. Despite significant architectural advances in multimodal agentic systems, our comprehensive analysis reveals several fundamental limitations that persist across application domains and fusion strategies.

**The "Grounding Gap" in Complex Environments:** Despite the theoretical advantages of using early-fusion models in agentic frameworks, as demonstrated by us in Section 5, there is a persistent performance gap that remains between agents and humans in high-precision domains like GUI navigation. We have shown this gap to be ∼60-70% on benchmarks like WebArena (Zhou et al., 2024b) and VisualWebArena (Koh et al., 2024) in Section 5.2. This suggests that while native perception improves semantic understanding, current architectures still lack precise coordinate-level grounding and multi-step error-recovery mechanisms necessary for reliable interactive tasks.

**The Unresolved Performance-Efficiency Trade-off:** Our efficiency analysis in Section 6 has exposed a critical dilemma. While native multimodal agents such as GPT-4V/o (OpenAI, 2023; 2024) and Gemini (Gemini Team, 2024) exhibit strong generalization and superior zero-shot capabilities, they incur prohibitive inference costs and latency at scale. On the other hand, fine-tuned domain-specific models achieve higher efficiency but require substantial upfront training compute and exhibit limited generalization. Our survey indicates that no single architectural approach successfully balances state-of-the-art performance with deployment feasibility.

**Brittleness of Memory and Reasoning over Long-Horizon:** Although frameworks like Reflexion (Shinn et al., 2023b) and HM-RAG (Zhang et al., 2025b) demonstrate episodic memory capabilities, our analysis reveals that current memory architectures struggle with error accumulation, belief revision, and consistent state maintenance over extended interactions. In long-form video understanding applications (Section 4.5, even retrieval-based approaches that process only task-relevant frames face challenges in fine-grained reasoning over hour-scale videos. This is due to the absence of robust mechanisms for forgetting outdated information and maintaining temporal coherence in tasks requiring sustained context.

**Performance Verification and Benchmarking Leakage:** A substantial fraction of state-of-the-art results across all application domains are coming from frameworks that rely solely on proprietary, closed-source API models like GPT-4V/o (OpenAI, 2023; 2024), Gemini (Gemini Team, 2024) for orchestration. This prevents independent verification of reported gains, systematic ablation studies, and understanding of failure modes of these frameworks. This limits scientific progress in agentic research within the open community. Furthermore, reliance on publicly available internet data for benchmarks such as WebArena (Zhou et al., 2024b), VisualWebArena (Koh et al., 2024), and Video-MME (Fu et al., 2024) may lead to performance inflation due to potential data leakage. This makes it difficult to distinguish the agentic framework's generalization capabilities from model memorization.

**Lack of Adversarial Robustness:** The inclusion of richer sensory inputs significantly expands the attack surface of multimodal architectures. In digital settings, adversarial visual elements can function as covert prompt injections, silently hijacking an agent's control flow without explicit textual manipulation (Foerster et al., 2026). In physical and embodied environments, these vulnerabilities become even more critical due to cross-layer threats. Subtle visual perturbations can seamlessly mislead lower-level perception systems, propagating errors upward into high-level cognitive decision-making. For instance, in autonomous driving scenarios, these adversarial distortions can directly cause agents to adopt fundamentally unsafe planning behaviors (Eslami & Yu, 2025).

**Safety-Critical Failure Modes:** Beyond malicious attacks, these systems exhibit intrinsic operational fragilities, primarily driven by weak multimodal grounding and a fundamental lack of foresight. Agents are frequently distracted by irrelevant or spurious visual features, leading to instruction drift and hallucinated actions (Ma et al., 2024a). Although recent approaches like Hydra attempt to mitigate this through online self-correction (Jalaian et al., 2025), such mechanisms remain brittle. Consequently, empirical benchmarks demonstrate that embodied agents frequently fail to navigate physical hazards safely, particularly in complex, long-horizon, and dynamically evolving tasks (Ying et al., 2025; Chen et al., 2026a). This fragility is severely compounded by the fact that current architectures operate largely reactively as they lack robust mechanisms, such as predictive world models, to simulate and evaluate the downstream, safety-critical consequences of their actions before execution (Chen et al., 2026b).

**Hallucination in Multimodal Grounding and Reasoning:** A fundamental limitation of current frameworks is the tendency to *hallucinate* actions based on flawed perception or distorted logic. Environmental distractions such as irrelevant visual background features or spurious UI elements frequently cause agents to drift from their original instructions and execute unintended, often hazardous, actions (Ma et al., 2024a). More critically, even when perception is accurate, the reasoning backbone may hallucinate physical affordances or fabricate invalid planning steps that do not exist in the environment. While emerging architectures like Hydra attempt to catch these errors through online self-correction and cross-model verification (Jalaian et al., 2025), these mechanisms remain brittle in unconstrained settings.

## 11 Future Research Directions

To address the persistent gaps in grounding, memory, and efficiency identified in our analysis, we highlight several research trajectories to help shape the next generation of multimodal agents.

**Unified Multimodal Representation and Reasoning:** To bridge the gap between rich perception and reliable actions, future research must move beyond treating multimodal fusion as an embedding problem and instead develop representation spaces that are explicitly optimized for downstream planning and control. Promising directions include attention mechanisms that dynamically reweight modalities based on task context, intermediate representations that preserve spatial and temporal structure while remaining compatible with symbolic reasoning, and theoretical models that formalize how perceptual evidence should influence planning and tool selection within the agentic loop.

**Scalable Memory Architectures for Lifelong Learning:** Unlike current systems (Shinn et al., 2023b; Zhang et al., 2025b), which rely on passive storage, future agentic frameworks require memory architectures inspired by cognitive science that can manage multiple timescales of knowledge. More importantly, these memory systems should be able to revise their existing beliefs by identifying conflicting knowledge based on

new evidence, tracing these conflicts to their sources, and updating their beliefs coherently. This requires moving beyond retrieval-augmented generation toward memory systems that actively reason about their own knowledge state.

**Hybrid Architectures for Resolving Trade-offs:** As we have seen in Section 10, both large natively multimodal models and fine-tuned domain-specific models have their strengths and weaknesses. Therefore, future work should explore hybrid architectures that strategically combine the strengths of both systems. One direction could be *hierarchical systems*, inspired by (Gupta et al., 2025a), in which lightweight models handle routine tasks while larger models address edge cases, thereby significantly reducing API costs. We could also use *distillation frameworks* to transfer the multimodal reasoning capabilities of proprietary models into smaller, locally deployable architectures through carefully designed intermediate supervision. Finally, Mixture-of-Expert designs could help reduce computational overhead without sacrificing performance by activating only the task-relevant perception and reasoning modules.

**Multimodal Integration Beyond Vision-Language:** Our survey reveals that, despite the multimodal label, current agents overwhelmingly focus on vision-language fusion while neglecting other sensory modalities critical for embodied intelligence. Future work must extend to domains such as *tactile and force sensing* in robotic manipulation, where contact dynamics provide information unavailable through vision alone. *3D spatial reasoning* is also essential, as it imparts volumetric knowledge rather than projecting 3D scenes onto 2D images, which are required for navigation and manipulation planning. Researchers must also address *cross-modal reasoning*, where information from one modality can resolve ambiguity in another, as shown in (Galougah et al., 2025b).

**Multimodal Multi-Agent Communication:** An important emerging frontier is the development of native multimodal communication protocols for multi-agent systems, moving beyond the current reliance on natural language as the primary medium of coordination. While language provides a flexible interface, it introduces latency, ambiguity, and information loss when encoding dense modalities such as vision, spatial layouts, or embodied states. Recent works have begun to explore alternative paradigms, including latent-space communication (Zou et al., 2025; Yu et al., 2026b), where agents exchange continuous hidden representations, and unified latent alignment mechanisms such as the Universal Visual Codec (Liu et al., 2026). In parallel, structured spatial protocols have been investigated for grounding coordination in explicit geometric representations (Padhan et al., 2026), while program-level embodied communication enables agents to share executable plans and trajectories for safe physical coordination (Shi et al., 2026). Despite promising early results, these approaches face significant challenges in interoperability across heterogeneous architectures, interpretability of exchange signals, and robustness in safety-critical settings. Future research should focus on designing standardized, efficient, and verifiable communication interfaces that enable scalable coordination while preserving the richness of multimodal information.

## 12 Conclusion

Across domains, the evidence indicates a consistent pattern: the largest gains in multimodal agents arise from tighter integration of perception and reasoning, rather than from scaling model size alone. In robotics, compact end-to-end systems can outperform substantially larger pipelines that separate perception from action. In complex digital environments, such as web navigation, current agents remain far from human-level reliability, particularly for tasks requiring precise control and error recovery. These findings also carry clear implications for efficiency and deployment. While native large multimodal models provide strong zero-shot capabilities, their reliance on token-heavy inference introduces latency and cost that limit practical scalability. In contrast, domain-specific and fine-tuned architectures offer a more sustainable trade-off. Systems such as OpenVLA (Kim et al., 2024) and CogAgent (Hong et al., 2024) illustrate that higher training costs can be amortized through lower inference latency and improved long-term efficiency. Similar principles apply in high-bandwidth settings, such as long-form video generation, where retrieval-based designs like VideoAgent (Wang et al., 2024e) and VLog (Lin & Shou, 2025), along with frozen backbones, reduce redundant computation. Overall, this suggests that future progress will depend less on building ever-larger generalist models and more on efficient, grounded architectures tailored to interactive tasks, particularly in settings where reliability and responsiveness are critical.

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
