# OpenReview forum: "A Survey on Foundations and Frontiers of Multimodal Agentic Frameworks: Techniques and Applications"
_TMLR — Accepted by TMLR_

### Review · Reviewer_bdQ3 · 2026-03-10

**Summary Of Contributions:**

This comprehensive survey explores the landscape of multimodal agentic frameworks by tracing their evolution from traditional text-centric models to advanced systems capable of integrating diverse modalities such as images, audio, and video. The authors establish a systematic taxonomy categorized by perception fusion strategies while analyzing how multimodality transforms core functional modules including reasoning, planning, memory, and action across various domains like robotics and web navigation. By providing a detailed evaluation of performance, efficiency, and scalability trade-offs, the paper identifies critical architectural gaps and charts a strategic roadmap for the development of robust, general-purpose intelligent agents.

**Audience:**

Yes

**Audience Explanation:**

The paper's insights into perception fusion strategies and cross-domain scalability offer a valuable reference for those designing the next generation of robust, general-purpose intelligent agents.

**Broader Impact Concerns:**

As this work is a systematic survey of existing literature and does not introduce novel data collection involving human subjects or high-risk autonomous deployments, there are no specific ethical concerns or broader impact implications that require additional disclosure.

**Claims And Evidence:**

Yes

**Claims Explanation:**

The submission provides a rigorous and comprehensive taxonomy of multimodal agents, supported by a systematic synthesis of state-of-the-art literature and empirical performance benchmarks across diverse application domains. The comparative analysis of perception fusion strategies and efficiency trade-offs is backed by clear technical definitions and extensive evidence from numerous cited works, making the core claims highly convincing.

**Requested Changes:**

1. The performance analysis section would benefit from more standardized, side-by-side comparison tables to facilitate direct quantitative comparisons between different agentic frameworks across the same benchmarks.
2. The technical discussion on multimodal memory architectures could be expanded to provide more specific details on how various systems manage state-space complexity during prolonged, high-bandwidth interactions.
3. While the taxonomy of fusion strategies is well-defined, the survey could offer a more critical analysis of the specific software engineering challenges and deployment constraints associated with integrating these complex modules into production-ready pipelines.
4. The paper's treatment of evaluation methodologies would be more impactful if it included a deeper investigation into the systemic biases and limitations of current LLM-based evaluators when assessing multimodal agent performance.
5. The section on future research directions could more explicitly address the nascent field of multi-agent collaboration within multimodal environments, specifically focusing on communication protocols for non-textual information exchange.

---

> ### Author Response · Authors · 2026-04-14
> **Authors' Response - (1/3)**
>
> We thank the reviewer for their detailed comments. We have highlighted all corresponding changes in the revised manuscript in $\color{violet}\textbf{VIOLET}$.
>
> > The performance analysis section would benefit from more standardized, side-by-side comparison tables to facilitate direct quantitative comparisons between different agentic frameworks across the same benchmarks.
>
> We would like to draw the reviewer's attention to Section-5, where we have explained how performance changes across different frameworks. However, we agree that a tabular, analytical representation of the same would be much easier to comprehend. Based on this suggestion, we are sharing the following four tables, which analyse the performance of representative frameworks across common benchmarks. We have added these tables to the main paper, under the respective application areas in Section-5 (Pages 29-32), to make the performance comparison more intuitive.
>
> ---
>
> ### Robotics:
>
> | Benchmarks | Octo (93M) *by OpenVLA* | Octo (93M) *by Octo* | RT-2-X (55B) | OpenVLA (7B) |
> | :--- | :---: | :---: | :---: | :---: |
> | **Google Robot** | 26.7% | ~80% | 78.3% | 85.0% |
> | **BridgeData V2 WidowX** | 20% | ~50% | 50.6% | 70.6% |
> | **LIBERO Simulation** | 75.1% | - | - | 76.5% |
>
> OpenVLA has significantly outperformed RT-2-X (55B) on major evaluation benchmarks, suggesting that our conclusion about the efficiency of late-fusion architectures is correct. It is important to note that the benchmark scores of Octo vary significantly depending on the source. While OpenVLA has shown that Octo struggles with generalization on physical robots, Octo reports different scores, showing that their 93M-parameter generalist policy was able to match the performance of RT-2-X on the Google and WidowX robots.
>
> ---
>
> ### GUI & Web Navigation:
>
> | Benchmark | GPT-4 Baseline | WebAgent | AutoWebGLM | CogAgent | SeeClick |
> | :--- | :---: | :---: | :---: | :---: | :---: |
> | **Mind2Web** | 30.9% | 46.7% | 59.5% | 58.2% | 20.9% |
> | **MiniWob++** | 32.1% | 85.6% | 89.3% | - | 67.0% |
> | **AITW** | 50.54% | - | - | 76.88% | 66.4% |
> | **ScreenSpot** | 16.2% | - | - | 47.4% | 53.4% |
>
> Within the GUI and Web navigation space, frameworks are mostly specialized based on the application area. For web navigation, Text/HTML-based architectures such as AutoWebGLM and WebAgent perform better as they can leverage the underlying web code for high accuracy. However, GUI navigation is more complicated as the underlying code cannot be easily extracted, and graphical intricacies require visual perception. Therefore, lightweight fine-tuned models such as CogAgent and SeeClick are a preferred approach for building universal agents due to their visual grounding ability.
>
> ---
>
> ### Long-Form Video Understanding:
>
> | Benchmark | DoraemonGPT | VLog | LLoVi | VideoAgent |
> | :--- | :---: | :---: | :---: | :---: |
> | **EgoSchema** | - | 43.1% | 52.2% | 54.1% |
> | **NExT-QA** | 55.7% | - | 73.8% | 71.3% |
>
> Our analysis in this domain has shown that VideoAgent and LLoVi achieve the highest accuracy on standard video QA benchmarks. The reason becomes clear when we look at their approaches to managing contextual data without losing relevant information:
> - **LLoVi** achieves this through a multi-round summarization prompt that converts textual captions of video clips into a video summary using an LLM. The LLM's textual reasoning ability helps filter out noisy, redundant information.
> - **VideoAgent** uses an iterative frame selection process, where the LLM acts as an agent that evaluates its current knowledge, actively predicts what missing information it needs, and uses CLIP to retrieve only the most relevant frames from specific segments of the video.
>
> ---
>
> ### Multimedia Generation and Editing:
>
> The nature of this application area is quite subjective, and it becomes difficult to establish success and failure criteria for editing and generation tasks. Therefore, the evaluation of multimodal agents in this domain is frequently hindered by a lack of comprehensive, real-world benchmarks, which has prompted many researchers to manually curate their own datasets to test their specific frameworks and tasks. We have observed the following trends:
> - Most benchmarks lack realistic and multi-turn editing and generation samples and ground truths. While **MagicBrush** seems to be one of the very few large-scale benchmarks in this category, the authors have noted that most other datasets are either domain-specific or too noisy.
> - Standard benchmarks are insufficient for agentic systems handling complex, composite instructions. Therefore, frameworks such as **CoSTA*** and **FaSTA*** have developed custom benchmarks to evaluate cost-sensitive multimodal generation and editing. Frameworks like **VISPROG** and **CLOVA** were evaluated on standard benchmarks for compositional and grounded visual reasoning, like GQA, but for image editing, they had to collect their own instructions and images manually.

---

> ### Author Response · Authors · 2026-04-14
> **Authors' Response - (2/3)**
>
> > The technical discussion on multimodal memory architectures could be expanded to provide more specific details on how various systems manage state-space complexity during prolonged, high-bandwidth interactions.
>
> While our current classification of multimodal memory under Section 3.3 was based on a modality fusion perspective, we agree that the dynamic behaviour of different memory architectures can be better explained by classifying them based on (1) temporal pressure, where the memory state that must be maintained accumulates across steps and episodes, and (2) bandwidth pressure, where the information arriving at each timestep is dense, as is the case in continuous camera streams, full-page screenshots, or hour-scale videos. We have expanded the scope of Section 3.3 (Page 14) to include the above classification. The table below is a summary of our analysis.
>
> | State-Space Control | Management Strategy | Representative Frameworks | Mechanism |
> | :--- | :--- | :--- | :--- |
> | **Temporal Management** | Context Compression | Mobile-Agent-v2, Reflexion, MemGPT | Replaces exploding interaction history with concise text in terms of task progress or failure cases. |
> |  | Knowledge Retrieval | AppAgent-v2, HM-RAG, Voyager | Removes dependency on context length by building a static reference and retrieving only relevant chunks during inference. |
> |  | Truncation | Visual ChatGPT | Enforces a strict upper bound on visual context growth by aggressively pruning interaction history. |
> | **Bandwidth Management** | Structured Queries | VideoAgent, DoraemonGPT | Avoids dense sampling by querying a CLIP embedding index and retrieving only highly relevant frames. |
> |  | Modality Translation | LLoVi, Video-RAG | Offloads visual reasoning by sampling and captioning a sparse set of interaction frames. |
> |  | Generative Retrieval | VLog | Translates video content into a narration vocabulary and uses generative retrieval to search through it. |
>
> > While the taxonomy of fusion strategies is well-defined, the survey could offer a more critical analysis of the specific software engineering challenges and deployment constraints associated with integrating these complex modules into production-ready pipelines.
>
> Sections 6 and 7 of our paper actively evaluate inference latency, throughput, memory overhead, and API costs. To address the reviewer's feedback and make these software engineering challenges more explicit, we have introduced Latency and Reliability, in Sections 8 and 9 (Pages 39-42) respectively, as two formal evaluation axes in the revised manuscript. As summarized in the table below, framework designers currently face a strict trade-off: dense visual verification improves error recovery but degrades throughput, while reducing verification frequency introduces unacceptable failure rates in complex environments.
>
> | Deployment Axis | Framework | Bottleneck Addressed | Optimization Strategy | Production Impact |
> | :--- | :--- | :--- | :--- | :--- |
> | **Latency** | AutoDroid | Token processing overhead | Prunes UI by merging functionally equivalent elements. | Reduces token count by nearly 50%, accelerating inference. |
> |  | WebGLM | Sequential browsing delays | Employs an LLM-augmented retriever (RAG). | Achieves ~5.36s response times (nearly 10× faster than baseline WebGPT). |
> |  | CogAgent | High-res image processing | Processes 1120x1120 images with linear FLOP scaling via parallel cross-modules. | Makes high-resolution GUI processing computationally feasible and faster. |
> |  | Mobile-Agent-v2 | Long-context history limits | Multi-agent design where a planner condenses history into text summaries. | Speeds up decision-making compared to full interleaved history processing. |
> |  | EMMA | Explicit reasoning delays | Optimizes the end-to-end model to remove explicit reasoning chains. | Achieves onboard inference speeds of ~3 Hz. |
> |  | OpenVLA | VLA model parameter bulk | Uses a smaller 7B backbone and 4-bit quantization. | Achieves ~6 Hz throughput on a single consumer-grade GPU. |
> | **Reliability** | SayCan | Unsafe physical execution | Evaluates dual probabilities: semantic validity (language) and physical feasibility (affordance). | Ensures plans are physically executable before reaching actuators. |
> |  | Inner Monologue | Execution failure | Processes environmental observations (action success) to dynamically replan. | Enables closed-loop feedback and error recovery. |
> |  | COME-robot | Subtask failures | Employs a dual-loop mechanism (inner verification, outer replanning) via GPT-4V. | Significantly higher success rates in open-ended physical tasks. |
> |  | Mobile-Agent | Navigation errors | Self-reflection module triggers if visual state changes do not match expectations. | Allows the LMM to autonomously correct its own errors. |
> |  | AppAgent v2 | Zero-shot hallucination | Uses RAG to consult a reference document built during an exploration phase. | Improves memory retrieval efficiency and task reliability. |

---

> ### Author Response · Authors · 2026-04-14
> **Authors' Response - (3/3)**
>
> > The paper's treatment of evaluation methodologies would be more impactful if it included a deeper investigation into the systemic biases and limitations of current LLM-based evaluators when assessing multimodal agent performance.
>
> We appreciate the reviewer's attention to the potential biases of LLM-based evaluators. To address this comment systematically, we conducted an audit of the evaluation methodologies employed across all representative frameworks surveyed in our paper. We organize our findings in a new summary table that classifies evaluation approaches into five categories: (1) environment-based task completion, (2) programmatic functional correctness, (3) deterministic automated metrics, (4) human evaluation, and (5) LLM-as-a-judge. Our analysis shows that LLM-as-a-Judge evaluation accounts for only ~5% of all surveyed frameworks, confined exclusively to the GUI and web navigation domain.
>
> Therefore, we agree with the reviewer that LLM-based evaluators carry known systemic biases, including position bias, verbosity bias, and self-enhancement bias. However, given that only 5% of the frameworks in our survey use such evaluators, and those that do employ them primarily for narrow tasks like semantic equivalence checking, the overall impact on the reported performance landscape is limited. We have added the table below to the revised manuscript as a new Section 3.5 (Pages 15-17) under Taxonomy, to make this distribution explicit and include a brief discussion of these biases in the context of the two frameworks that employ LLM-based evaluation.
>
> | Application Domain | Evaluation Category | Primary Metric(s) | Representative Frameworks |
> | :--- | :--- | :--- | :--- |
> | **Robotics & Physical Embodiment** | Sim/Real-World Task Completion | Task success rate, planning success rate | SayCan, Inner Monologue, Code as Policies, PaLM-E, RT-2, OpenVLA, Octo, Magma |
> |  | Domain-Specific Driving Metrics | L2 distance, collision rate | EMMA |
> | **GUI & Web Navigation** | Programmatic Functional Correctness | State validation, exact/substring match | WebAgent, AutoDroid, AssistGUI, AutoWebGLM, SeeAct |
> |  | Deterministic Benchmark Accuracy | Element accuracy, grounding accuracy, step success rate | CogAgent, SeeClick, GUI-Owl, Ferret-UI, Pix2Act, WebGUM |
> |  | Human Evaluation | Human preference, task completion | WebGPT, AppAgent, Mobile-Agent, MM-Navigator |
> |  | LLM-as-a-Judge | GPT-4V trajectory evaluation; GPT-4 fuzzy matching (partial) | WebVoyager; WebArena (info-seeking subset only) |
> | **Long-Form Video Understanding** | Deterministic Automated Metrics | MCQ accuracy, temporal IoU, retrieval metrics | VideoAgent, LLoVi, DoraemonGPT, VideoMind, VLog |
> | **Multimedia Content Generation & Editing** | Deterministic Benchmark Scores | T2I-CompBench, FID, CLIP Score, objective audio metrics | VISPROG, GenArtist, CoSTA*, FaSTA*, NExT-GPT |
> |  | Human Evaluation | MOS, human preference studies | AudioGPT, WavCraft, WavJourney, LAVE, CLOVA |
>
> > The section on future research directions could more explicitly address the nascent field of multi-agent collaboration within multimodal environments, specifically focusing on communication protocols for non-textual information exchange.
>
> Native multimodal communication is still a huge bottleneck for multi-agent systems. When agents have to translate complex spatial reasoning into natural language, the latency and information loss are just too high. Since there is not yet a consensus on how to address this, we agree it is important to map out the current experimental landscape. To address this, we have surveyed current literature on this topic and proposed a taxonomy that groups these new non-textual protocols into three main clusters. This helps clarify the trade-offs researchers are currently grappling with. We have added our analysis under Section-11 (Page-44) as an area that requires further attention from the community.
>
> | Protocol Cluster | Core Mechanism | Primary Application | Example Frameworks |
> | :--- | :--- | :--- | :--- |
> | **Deep Latent** | Sharing intermediate feature tensors, continuous embeddings, and cross-attention maps. | Preserving raw, uncompressed visual context without tokenization loss. | LatentMAS (2025), L2-VMAS (2026), Universal Visual Codecs |
> | **Spatial & Grounding** | Exchanging structured likelihoods, spatial kernels, and geometric coordinates. | Collaborative object detection, spatial anchoring, and allocentric mapping. | MAPG (2026) |
> | **Embodied & Kinematic** | Broadcasting raw action vectors, executable path programs, and state trajectories. | Physical coordination and collision-free pathing in dynamic environments. | CaPE (2026) |

---

> > ### Author Response · Authors · 2026-04-24
> >
> > Dear Reviewer,
> >
> > We are thankful for your valuable feedback. We have tried to address all of your concerns in our responses point by point. As the author-reviewer discussion period will end soon (April 28), we would love to hear if you still have any concerns, and we are more than happy to discuss them.
> >
> > We are looking forward to your comments.

---

### Review · Reviewer_TY46 · 2026-03-29

**Summary Of Contributions:**

The submission provides a comprehensive survey on the evolution and architecture of multimodal agentic frameworks.
The authors introduce a modality-centric taxonomy that categorizes how sensory data is integrated into agentic systems: delegated perception (via tool calling), late-fusion perception, and early-fusion (native) perception.
Using this taxonomy, the paper analyzes the impact of multimodality across core functional modules, including reasoning, planning, memory, and action spaces.
The authors evaluate these frameworks across four primary application domains: Robotics, GUI & Web Navigation, Multimedia Content Generation & Editing, and Long-form Video Understanding.
Finally, the paper provides a quantitative and qualitative mapping of performance, efficiency (latency/throughput), and scalability (compute/cost) trade-offs to guide future architectural choices.

**Audience:**

Yes

**Audience Explanation:**

TMLR's readership includes researchers and practitioners focused on foundation models, embodied AI, and complex machine learning systems.
A unified analysis that connects foundational architectural choices, such as modality fusion strategies and memory mechanisms, to practical capabilities and deployment constraints is highly relevant.
Engineers and applied researchers will be particularly interested in the paper's roadmap, which highlights the latency and economic advantages of domain-specific architectures over heavy, proprietary APIs.

**Broader Impact Concerns:**

None.

**Claims And Evidence:**

Yes

**Claims Explanation:**

The authors systematically support their architectural claims by citing specific performance metrics from established benchmarks across the surveyed domains.
For instance, to support the claim that architectural integration is more important than raw parameter scale in embodied agents, they cite that the 7B OpenVLA model outperforms the 55B RT-2-X by 16.5%.
Similarly, the authors back their claims about the persistent "Grounding Gap" in GUI navigation by highlighting the 60-70 percentage point disparity between GPT-4V-based agents and human performance on benchmarks like WebArena and VisualWebArena.
The arguments regarding efficiency and scalability are also well-supported by contrasting the high inference latency and recurring token costs of API-based models against the upfront compute costs of fine-tuned late-fusion models.

**Requested Changes:**

Include More Recent State-of-the-Art Models. The paper is already highly comprehensive, offering detailed introductions and comparisons across multiple downstream agent domains. However, given the exceptionally rapid pace of advancement in multimodal large models and their downstream agentic frameworks, incorporating and comparing more recent cutting-edge models would further elevate the quality, timeliness, and relevance of this work.

---

> ### Author Response · Authors · 2026-04-14
>
> > Include More Recent State-of-the-Art Models. The paper is already highly comprehensive, offering detailed introductions and comparisons across multiple downstream agent domains. However, given the exceptionally rapid pace of advancement in multimodal large models and their downstream agentic frameworks, incorporating and comparing more recent cutting-edge models would further elevate the quality, timeliness, and relevance of this work.
>
> We thank the reviewer for this suggestion. We have incorporated fifteen recent state-of-the-art frameworks across all four application domains, reflecting rapid advancements from late 2024 through early 2026:
>
> **Robotics and Physical Embodiment (Section 4.2, Page-20)**:
> $\pi_0 $​[1], $\pi_{0.5}$[2], VTAM [3], GR00T N1 [4], CoT-VLA [5], and HALO[6].
>
> **GUI and Web Navigation (Section 4.3, Page 22-23):**
> UI-TARS-2 [7], Agent S2 [8], OpenCUA [9], InReAct [10], and GUI-Genesis [11].
>
> **Multimedia Content Generation and Editing (Section 4.4, Page-25):**
> CREA [12]. This is in $\color{orange}\textbf{ORANGE}$ as it is common with suggestions made by Reviewer enSS.
>
> **Long-Form Video Understanding and Retrieval (Section 4.5, Page 27-28):**
> VideoRAG [13], VideoDeepResearch [14], and Deep Video Discovery (DVD) [15].
>
> For each framework, we have added detailed descriptions and analysis in the corresponding application sections. These additions capture critical recent architectural trends, including continuous flow-matching and multimodal chain-of-thought reasoning for physical manipulation (π0\pi_0 π0​, π0.5\pi_{0.5}  π0.5​, CoT-VLA, HALO), multi-turn reinforcement learning and code-native reward synthesis for precise GUI grounding (UI-TARS-2, InReAct, GUI-Genesis), multi-agent collaborative creativity (CREA), and advanced orchestrated discovery mechanisms over dense context-window scaling for unlimited-length videos (VideoRAG, DVD).
> All new additions have been highlighted in $\color{red}\textbf{RED}$ in the revised manuscript to facilitate your review.
>
> ---
>
> ### References:
> [1] Black, Kevin, et al. "$\pi_0 $: A Vision-Language-Action Flow Model for General Robot Control." arXiv preprint arXiv:2410.24164 (2024).
> [2] Intelligence, Physical, et al. "$\pi_ {0.5} $: a Vision-Language-Action Model with Open-World Generalization." arXiv preprint arXiv:2504.16054 (2025).
> [3] Yuan, Haoran, et al. "VTAM: Video-Tactile-Action Models for Complex Physical Interaction Beyond VLAs." arXiv preprint arXiv:2603.23481 (2026).
> [4] GR00T Team, "GR00T N1: An Open Foundation Model for Generalist Humanoid Robots," arXiv:2503.14734, 2025.
> [5] Zhao et al., "CoT-VLA: Visual Chain-of-Thought Reasoning for Vision-Language-Action Models," CVPR 2025, arXiv:2503.22020.
> [6] Shou et al., "HALO: A Unified Vision-Language-Action Model for Embodied Multimodal Chain-of-Thought Reasoning," arXiv:2602.21157, 2026.
> [7] Wang et al., "UI-TARS-2: Advancing GUI Agent with Multi-Turn Reinforcement Learning," arXiv:2509.02544, 2025.
> [8] Agashe et al., "Agent S2: A Compositional Generalist-Specialist Framework for Computer Use Agents," arXiv:2504.00906, 2025.
> [9] Wang et al., "OpenCUA: Open Foundations for Computer-Use Agents," NeurIPS 2025 Spotlight, arXiv:2508.09123.
> [10] Wang et al., "InReAct: An Inspire-Then-Reinforce Training Framework for Multimodal GUI Agent," EMNLP 2025 Findings.
> [11] Yu et al., "GUI-Genesis: Automated Synthesis of Efficient Environments with Verifiable Rewards for GUI Agent Post-Training," arXiv:2602.14093, 2026.
> [12] Venkatesh et al., "CREA: A Collaborative Multi-Agent Framework for Creative Image Editing and Generation," arXiv:2504.05306, 2025.
> [13] Ren et al., "VideoRAG: Retrieval-Augmented Generation with Extreme Long-Context Videos," arXiv:2502.01549, 2025.
> [14] Yuan et al., "VideoDeepResearch: Long Video Understanding with Agentic Tool Using," arXiv:2506.10821, 2025.
> [15] Zhang et al., "Deep Video Discovery: Agentic Search with Tool Use for Long-form Video Understanding," NeurIPS 2025, arXiv:2505.18079.

---

> > ### Author Response · Authors · 2026-04-24
> >
> > Dear Reviewer,
> >
> > We are thankful for your valuable feedback. We have tried to address all of your concerns in our responses point by point. As the author-reviewer discussion period will end soon (April 28), we would love to hear if you still have any concerns, and we are more than happy to discuss them.
> >
> > We are looking forward to your comments.

---

### Review · Reviewer_enSS · 2026-03-31

**Summary Of Contributions:**

This survey examines how multimodality has transformed agentic frameworks built around large language and multimodal models. The paper proposes a modality-centric taxonomy that organizes agentic systems along their perception, reasoning, planning, memory, and action modules. A key contribution is the formalization of three perception fusion strategies: delegated, late-fusion, and early-fusion. The survey covers four application domains including robotics, GUI/web navigation, multimedia content generation, and long-form video understanding, and provides cross-domain analysis of performance, efficiency, and scalability trade-offs.

**Strengths**
- The perception fusion taxonomy is a clean and useful organizing principle.
- The paper connects architectural choices to quantitative performance outcomes across domains.
- The efficiency and scalability sections offer practical guidance that is often missing from survey papers, including concrete numbers on inference latency, training costs, and deployment trade-offs.
- The coverage is broad and reasonably up to date, spanning work from 2021 through early 2025.

**Weaknesses**
- The survey is heavily descriptive in many sections, reading more like a structured enumeration of papers than a critical synthesis with novel insights.
- The discussion of multi-agent systems is thin given their growing importance in agentic AI.

**Audience:**

Yes

**Audience Explanation:**

The paper is relevant to people interested in the intersection of multimodality and agentic AI.

**Claims And Evidence:**

Yes

**Claims Explanation:**

The survey is mostly comprehensive, covering the current landscapes of multimodel agents.

**Requested Changes:**

1. Several sections, particularly 4.1 through 4.5, read as sequential paper summaries rather than synthesized analysis. For each application domain, the authors should more explicitly articulate what lessons emerge from comparing systems within that domain rather than describing each framework in isolation. The current structure often feels like "Framework X does A, Framework Y does B" without enough connective analysis explaining why certain design choices succeed or fail relative to others.
2. The paper lacks a dedicated discussion of safety, robustness, and failure modes of multimodal agents. Given the real-world deployment scenarios discussed, including robotics and autonomous driving, this omission is notable. Even a brief subsection in the Limitations or Future Directions addressing adversarial robustness, hallucination in multimodal grounding, and safety-critical failure patterns would strengthen the paper.

---

> ### Author Response · Authors · 2026-04-14
>
> We are grateful to the reviewer for highlighting important points that have helped us improve the quality of our survey significantly. To facilitate the review process, we have highlighted all corresponding changes in the revised manuscript in $\color{orange}\textbf{ORANGE}$.
>
> > The discussion of multi-agent systems is thin given their growing importance in agentic AI.
>
> We thank the reviewer for bringing up an important aspect of research in multimodal agentic systems. We agree with the reviewer that multi-agent systems are quite important in the current landscape of agentic systems. While we had not talked about them extensively in our current draft, as it was out of our initial scope, we have discussed them in detail in Section 4 (Pages 20-28) separately under each application domain. Some of the additions are common with suggestions made by Reviewer TY46, and are hence highlighted in $\color{red}\textbf{RED}$.
>
> > The survey is heavily descriptive in many sections, reading more like a structured enumeration of papers than a critical synthesis with novel insights.
>
> > Several sections, particularly 4.1 through 4.5, read as sequential paper summaries rather than synthesized analysis. For each application domain, the authors should more explicitly articulate what lessons emerge from comparing systems within that domain rather than describing each framework in isolation. The current structure often feels like "Framework X does A, Framework Y does B" without enough connective analysis explaining why certain design choices succeed or fail relative to others.
>
> We appreciate the reviewer's feedback on the writing style followed in Section-4 of the paper. Per the reviewer’s suggestion, we have added concise takeaway paragraphs to each subsection under Section-4 (Pages 17-28) to clearly distill the key insights from our analysis. We hope these additions improve the readability of the paper and make the main outcomes of each category of frameworks easier to grasp.
>
> > The paper lacks a dedicated discussion of safety, robustness, and failure modes of multimodal agents. Given the real-world deployment scenarios discussed, including robotics and autonomous driving, this omission is notable. Even a brief subsection in the Limitations or Future Directions addressing adversarial robustness, hallucination in multimodal grounding, and safety-critical failure patterns would strengthen the paper.
>
> We thank the reviewer for highlighting this important aspect of the analysis of agentic systems. While the manuscript already discusses deployment-related trade-offs in the sections on latency, reliability, and limitations, we agree that safety and failure modes deserve more explicit treatment. We have therefore added a dedicated discussion on adversarial robustness, hallucination in multimodal grounding, and safety-critical failure patterns under Section-10 (Limitations, Pages 42-43) of the revised manuscript. This discussion draws on recent evidence that multimodal agents remain vulnerable to environmental distractions in GUI settings, prompt injection and control-flow attacks in computer-use agents, and hazardous-instruction failures in embodied settings. We further highlight that current systems remain reactive and would benefit from predictive world models and stronger safety-aware planning mechanisms.

---

> > ### Author Response · Authors · 2026-04-24
> >
> > Dear Reviewer,
> >
> > We are thankful for your valuable feedback. We have tried to address all of your concerns in our responses point by point. As the author-reviewer discussion period will end soon (April 28), we would love to hear if you still have any concerns, and we are more than happy to discuss them.
> >
> > We are looking forward to your comments.

---

### Author Response · Authors · 2026-04-14
**Summary of Additions**

Dear AE,

Thank you again for the smooth coordination and for handling the review process so efficiently.

We have now submitted our rebuttal and made a concerted effort to thoroughly address all reviewer comments, along with revising the manuscript accordingly. At a high level, we have made the following key changes:

* **Improved synthesis and clarity:** We revised the Application section to improve readability and understanding, by adding concise takeaway notes under each subsection, highlighting key insights and comparative lessons across domains.

* **Incorporated recent state-of-the-art works:** We added several recent frameworks (early 2025–2026) across all application domains to ensure the survey reflects the latest developments in multimodal agentic systems.

* **Broadened discussion of multi-agent systems and future directions:** We expanded the coverage of multi-agent systems across application domains due to their increasing importance and added a taxonomy of emerging non-textual communication protocols in multi-agent systems under future research directions.

* **Enhanced performance analysis:** We introduced standardized, side-by-side comparison tables across benchmarks under Performance Analysis, to improve clarity and facilitate direct quantitative comparisons.

* **Expanded discussion on memory and system dynamics:** We extended the Taxonomy section to include a more detailed analysis of temporal and bandwidth pressures in multimodal memory, along with corresponding management strategies.

* **Strengthened evaluation methodology analysis:** We added a structured taxonomy of evaluation approaches and discussed the limitations and biases of LLM-based evaluators, along with their relatively limited use in current frameworks.

* **Enhanced deployment-focused analysis:** We introduced latency and reliability as additional independent evaluation axes, highlighting practical trade-offs and engineering challenges in real-world deployment of multimodal agentic systems.

* **Expanded safety, robustness, and failure mode discussion:** We included a dedicated discussion in the Limitations section addressing adversarial robustness, hallucination in multimodal grounding, and safety-critical failures.

We believe these revisions have significantly improved the clarity, completeness, and critical depth of the survey, and we are grateful to the reviewers for their valuable feedback.

---

### Author Response · Authors · 2026-05-14

Dear AE,

Thank you for your seamless handling of the review process. We are encouraged by the reviewers’ positive feedback and would be happy to address any remaining concerns, if needed.

We wanted to kindly follow up on the current status of our submission. Please let us know if there is anything further required from our side.

Thank you again!

---

### Decision · Action_Editor_SgY1 · 2026-05-13

**Recommendation:** Accept as is

**Audience:**

Yes

**Audience Explanation:**

This paper should be of interest to at least some members of the TMLR audience. Multimodal agentic systems are an important research direction at the intersection of foundation models, embodied AI, GUI/web agents, long-form video understanding, and multimodal reasoning. The paper provides a unified synthesis of different modality fusion strategies, system architectures, performance characteristics, efficiency costs, deployment constraints, and future research directions. It can therefore serve as a valuable reference for researchers working on multimodal agents, embodied intelligence, complex machine learning systems, and applied foundation models.

**Claims And Evidence:**

Yes

**Claims Explanation:**

The main claims of this paper are supported by reasonably sufficient, clear, and credible evidence. The core contribution of the paper is a systematic synthesis of multimodal agentic frameworks, including a modality-centric taxonomy, different perception fusion strategies, performance comparisons across application domains, efficiency and scalability trade-offs, memory mechanisms, and deployment constraints. These claims are primarily supported by extensive relevant literature, benchmark results, and cross-domain comparisons. Although the initial submission had some weaknesses, such as insufficient synthesis and limited discussion of safety and failure modes, these issues appear to have been largely addressed during the rebuttal process. Therefore, I believe that the paper’s support for its claims satisfies the relevant TMLR criteria.